# DynamicInfer: Runtime-Aware Sparse Offloading for LLMs Inference on a Consumer-Grade GPU

**Zhui Zhu**[1,*] **Weichen Zhang**[2,*] **Zhenghan Zhou**[2], **Yunhao Liu**[1,†] **& Fan Dang**[3]
[1]Department of Automation and BNRist, Tsinghua University
[2]Global Innovation Exchange, Tsinghua University
[3]School of Software Engineering, Beijing Jiaotong University
`{z-zhu22, weic_zhang23, zhouzh23}@mails.tsinghua.edu.cn`
`yunhao@tsinghua.edu.cn, dangfan@bjtu.edu.cn`

## Abstract

Large Language Models (LLMs) have achieved remarkable success in various NLP tasks, but their enormous memory footprints pose significant challenges for deployment on consumer-grade GPUs. Prior solutions, such as PowerInfer, combine offloading and sparse activation to reduce memory and computational overhead, but suffer from static neuron partitioning, leading to suboptimal GPU utilization and increased latency. In this work, we present DynamicInfer, a runtime neuron offloading framework that dynamically adapts neuron scheduling based on input-dependent activation patterns. DynamicInfer introduces (1) a hierarchical neural caching strategies, (2) a load-aware neuron activation mechanism tailored to heterogeneous hardware, and (3) an activation-aware prefetching pipeline that overlaps data transfer with computation. Extensive experiments on ReluLLaMA and Prosparse models across multiple hardware platforms demonstrate that DynamicInfer achieves up to 253% speedup over llama.cpp and 59% over PowerInfer, while retaining model accuracy. Our approach offers a practical and scalable solution for high-performance LLM inference on resource-constrained devices.

## 1 Introduction

Large language models (LLMs) have revolutionized natural language processing tasks such as text summarization (Zhang et al., 2024; Shakil et al., 2024), natural language understanding (Yu et al., 2024; Zhao et al., 2023), and dialogue systems (Yi et al., 2024; Guan et al., 2025). While traditionally deployed in data centers, there is a rapidly growing interest in deploying LLM on personal devices, driven by increasing concerns over data privacy (by PrivateGPT, 2023), customization needs (Liu et al., 2024c; Xu et al., 2024), and inference cost reduction (Lyu et al., 2023).

However, deploying LLMs on consumer-grade hardware faces fundamental challenges. Modern LLMs contain tens or even hundreds of billions of parameters, requiring memory footprints far beyond the capacity of mainstream consumer-grade GPUs (e.g., RTX 4090 with 24GB memory).

*Model offloading* (Song et al., 2023; Gerganov, 2025; HuggingFace, 2025; Sheng et al., 2023) and *sparse activation* (Liu et al., 2023; 2024a; Wang et al., 2024; Shin et al., 2025) have emerged as promising solutions to address the memory constraints of LLM inference on consumer devices. Offloading stores part of the model weights in the relatively large CPU memory and incrementally prefetches them into GPU memory (Sheng et al., 2023; HuggingFace, 2025) or directly computes in CPU during inference (Gerganov, 2025; Song et al., 2023).

PowerInfer (Song et al., 2023) pioneers the idea of combining model offloading with sparse activation, enabling efficient LLM inference on personal devices. However, despite its improvements, we found that PowerInfer's static hot-cold neuron partitioning often leads to suboptimal inference

---

[*]Equal contribution.
[†]Corresponding author.

speed-ups (§3). This is because its one-time assignment is based on offline profiling of general activation trends, while actual active neurons can vary dynamically depending on the input prompt and decoding tokens. As a result, many active neurons end up being processed on the CPU, despite the GPU's higher computational capabilities, leading to increased latency and GPU under-utilization.

In this paper, we present DynamicInfer, a run-time neuron offloading framework for efficient LLM inference on consumer-grade GPUs. DynamicInfer introduces a lightweight online neuron scheduler that tracks the sparsity of activation during runtime and dynamically reallocates neurons between CPU and GPU based on their predicted activity. This online adaptation enables DynamicInfer to achieve better GPU utilization and lower latency compared to static offloading baselines such as PowerInfer.

However, building such a system entails overcoming two major challenges: (1) How to design an online scheduling strategy that maximizes GPU utilization without compromising inference accuracy? (2) How to construct a computation and data transfer pipeline tailored to the large weight footprint of FFN layers, enabling efficient scheduling such that dynamic adjustments incur minimal additional latency?

To address the aforementioned challenges, we propose the following three core mechanisms:

To accommodate the dynamic nature of sparse neuron activation during inference, we introduce a **hierarchical neural caching framework**. The system determines the allocation of neurons between GPU and CPU by utilizing predictors to forecast neuron activations (micro-level) in conjunction with historical activation data (macro-level).

Static sparsity thresholds often result in underutilized GPU neurons. To address this, we propose a **load-aware neuron activation mechanism** grounded in the principles of the sparsity predictor. This mechanism dynamically adjusts the activation thresholds for neurons on the GPU and CPU while adapting to the varying workload and preserving theoretical inference accuracy.

To mitigate the I/O bottlenecks caused by cross-device neuron migration, we propose **activate neuron prefetching**. Based on early-stage neuron activation predictions, the system preemptively plans the neuron migration required for the upcoming inference steps. By leveraging asynchronous transfer mechanisms, it pipelines the transfer of neuron weights with the execution of feedforward computations on both GPU and CPU.

Building on the aforementioned three core mechanisms, we propose DynamicInfer, a pioneering large-scale model inference framework that integrates online neuron scheduling with dynamic sparsity optimization. Compared to llama.cpp (Gerganov, 2025) and PowerInfer (Song et al., 2023), our method improves a generation speed (token/s) by 60%-253% and 11%-59% respectively in real-world testing, while simultaneously improving GPU utilization and overall efficiency.

The main contributions of this paper are as follows:

- We conduct a systematic study on the input-dependent and dynamic characteristics of sparse activation in the FFN layers of LLMs. We identify the fundamental limitations of offline scheduling under diverse input scenarios and propose a corresponding online neuron scheduling system.

- We design a hierarchical neural caching strategy and a load-aware neuron activation mechanism to significantly accelerate inference.

- By prefetching active neurons and introducing a pipelined scheduling mechanism that overlaps computation with neuron transfer, we further mitigate I/O bottlenecks and enable a pipelined acceleration of the inference process.

- We implement a prototype system, DynamicInfer, and evaluate it on RTX 4090 and RTX 2080Ti GPUs. Experimental results demonstrate that, while maintaining comparable inference accuracy, our system increases the generation speed by an average of 143.4% and 25.3% compared to existing solutions such as Llama.cpp (Gerganov, 2025) and PowerInfer (Song et al., 2023), respectively, and achieves up to 253.3% and 59.4% speedup in specific scenarios.

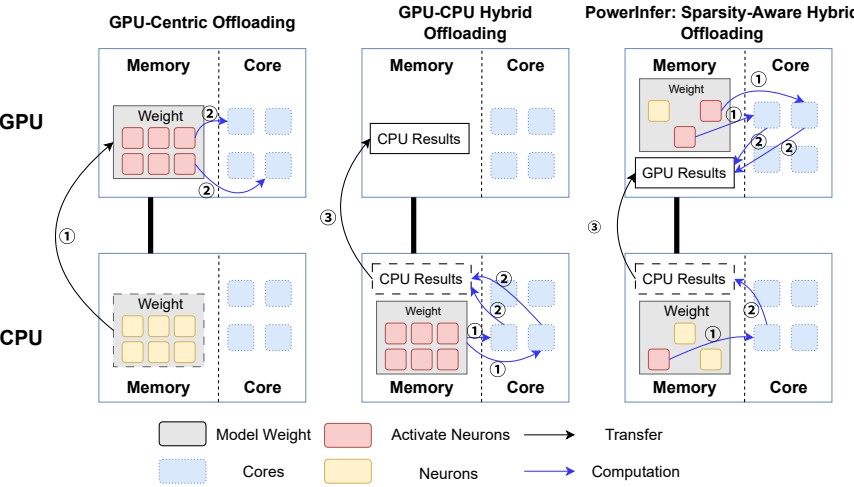

Figure 1: Offloading System

## 2 BACKGROUND

### 2.1 LLM OFFLOADING

Large language models typically consist of multiple transformer layers, with each layer composed of an attention component and a Feed-Forward Network (FFN) component (Touvron et al., 2023; Zhao et al., 2023). The majority of parameters in these large models are contributed by the FFN portion (Touvron et al., 2023).

As the size of LLMs continues to grow, deploying them on resource-constrained devices presents significant challenges, mainly because the consumer-grade GPUs on these personal devices lack sufficient memory capacity to accommodate the entire model (Song et al., 2023; Aminabadi et al., 2022).

Model offloading techniques offer a feasible solution to these challenges. *GPU-centric offloading methods*, such as DeepSpeed-Inference (Aminabadi et al., 2022) and FlexGen (Sheng et al., 2023), utilize huge CPU memory to store model parameters that exceed the GPU memory's capacity. During each inference iteration, computations are performed on the GPU, and parameters are transferred from the CPU memory as needed. These methods are well-suited for throughput-oriented scenarios, but the frequent data transfers over the PCIe bus between CPU and GPU introduce substantial latency overhead in consumer-grade GPU settings or under small-batch inference workloads.

*GPU-CPU hybrid offloading*, on the other hand, not only distributes model parameters across CPU and GPU memory, but also leverages both CPU and GPU processors to compute. For instance, in llama.cpp (Gerganov, 2025; Song et al., 2023), different model layers are allocated to either the CPU or GPU. The CPU handles computations for the first layers and passes intermediate activation values to the GPU for subsequent inference. This layered computation strategy effectively reduces both the frequency and volume of cross-device data transfers, thereby lowering inference latency. However, it introduces a new bottleneck: Since the computational performance (FLOPS) of CPUs is significantly lower than that of GPUs, the overall inference latency remains constrained.

## 3 OBSERVATIONS

Prior studies (Liu et al., 2023; 2024a; Wang et al., 2024; Song et al., 2023) show that many neurons in the self-attention and feed-forward network (FFN) modules are near-zero and have little impact on the final output. For instance, in ReLU-based models such as OPT and ProSparse, the FFN layers can achieve up to 90% sparsity (Liu et al., 2023; Song et al., 2023).

PowerInfer (Song et al., 2023) leverages this property by integrating sparse activation with model offloading to address memory constraints on personal devices. It starts with offline profiling of an entire training dataset to estimate each neuron's activation frequency. At runtime, PowerInfer uses

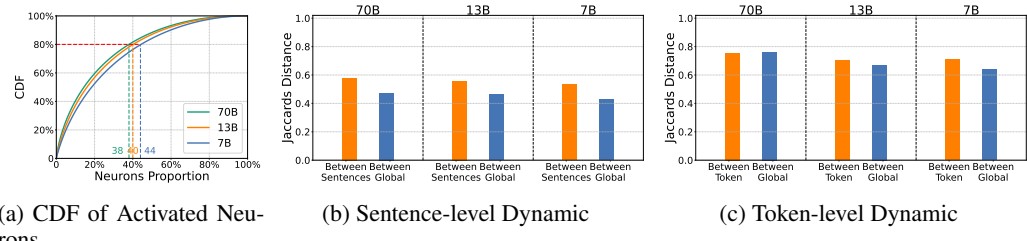

(a) CDF of Activated Neurons

(b) Sentence-level Dynamic

(c) Token-level Dynamic

Figure 2: Dynamics of Activated Neurons

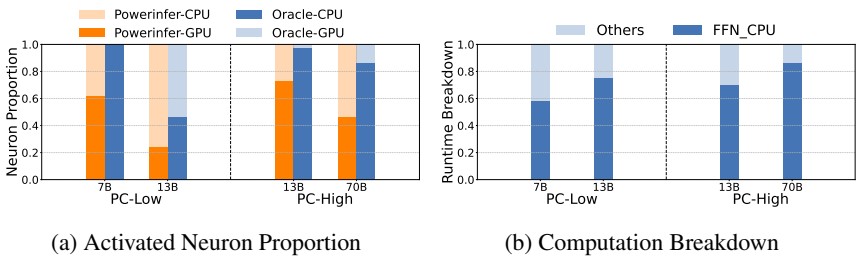

(a) Activated Neuron Proportion

(b) Computation Breakdown

Figure 3: Computation breakdown and neuron placement

sparse activation prediction to identify and compute only active neurons, skipping the inactive ones. Note that those infrequently (cold) neurons are still executed on the CPU once activated.

While PowerInfer's cold-hot neuron partitioning alleviates GPU memory constraints and reduces computational overhead, our observations reveal significant sentence-level and token-level variability in neuron activations, which undermines its effectiveness.

**Observation One: sentence-level neuron activation dynamics**. Similar to PowerInfer, when feeding LLMs different sentences (a.k.a., prompts), we observed that the activated neurons in FFN layers often exhibit locality where a small subset of neurons tends to be activated with high frequency figure 2a. However, this locality is highly input-dependent. Even if a set of high-frequency neurons is identified through offline profiling on the training data, the actual activation distribution can vary significantly for a new input.

Taking the ReluLLaMA models as examples, we randomly sampled 400 instances from the C4 dataset (Raffel et al., 2020). As illustrated in figure 2a, we observed that in specific layers, the top 40% of neurons account for approximately 80% of the activations. We define these as "hot neurons". However, considerable variation exists in hot neurons under different inputs. As shown in figure 2b, the Jaccard Distance between hot neuron sets for different inputs ranges from 0.53 to 0.57, indicating over 50% of hot neurons differ between inputs.

**Observation Two: token-level neuron activation dynamics**. In addition to the variation of sentence-level neuron locality, we also observed high variability of activated neurons between adjacent tokens within the same input sentence. As shown in figure 2c, we observed that the Jaccard Distance (Wikipadia) between the activated neurons of different tokens reaches values between 0.70 and 0.75, indicating substantial variation. Additionally, the Jaccard Distance between token-specific activated neurons and the globally defined hot neurons ranges from 0.64 to 0.76. This token-level dynamism exacerbates the inefficiency of global or offline neuron allocation strategies, as they are unable to adapt to the real-time computational demands of each inference step.

As a result, in PowerInfer, many hot neurons preloaded onto the GPU remain unused during inference, occupying valuable memory without contributing to computation. figure 3a illustrates that a considerable number of neurons are still computed on the CPU. In the figure, the Oracle represents the ratio of activated neurons to the number of neurons placed on the GPU, defined as #Activated Neurons / #GPU Neurons. Meanwhile, a bunch of cold neurons are occasionally activated and executed on the CPU (instead of the GPU), leading to considerable latency. For example, our benchmarks (see figure 3b) show that when running a 70B ReluLLaMA model on an RTX 4090 GPU (with 24GB memory), nearly 86% of the forward pass time is spent executing FFN computations on the CPU.

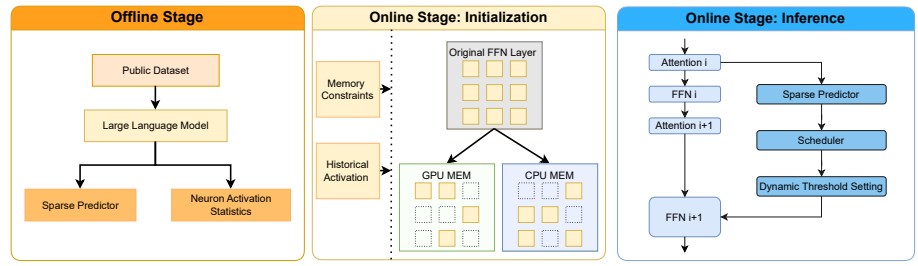

Figure 4: System Workflow

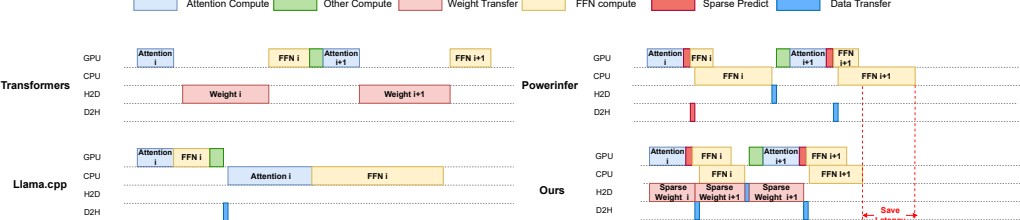

Figure 5: Compute Pipeline

# 4 SYSTEM DESIGN

## 4.1 DYNAMICINFER WORKFLOW

Our system operates in offline and online stages as figure 4.

**Offline Stage:** During the offline stage, we train the model's sparse predictor and simultaneously collect statistics on neuron activation frequencies. The sparse predictor is trained on the C4 dataset (Raffel et al., 2020) using 2,000 randomly sampled examples, generating approximately 700,000 output tokens.

**Online Stage:** During inference, based on the activation frequencies collected offline, we pre-load a subset of the most frequently activated (hot) neurons into GPU memory. This process is formulated as an optimization problem that aims to maximize the GPU cache hit rate while adhering to memory constraints.

During layer-by-layer inference, our framework uses the hidden representation of the current layer to asynchronously predict the activated neurons for the future layer via the sparse predictor and may prefetch the CPU neuron to the GPU. Activated neurons preloaded into GPU memory are processed on the GPU, while the remaining activated neurons are computed on the CPU, with their results transferred to the GPU for integration. Detailed allocation and loading implementation is provided in Appendix C.

## 4.2 NEURON I/O PIPELINE DESIGN

### 4.2.1 CROSS-LAYER SPARSITY PREDICTION

DynamicInfer reduces computation by leveraging sparse activation prediction. An MLP-based predictor takes the hidden state output of each attention layer as input and estimates which FFN neurons should be activated. During inference, the engine computes only those FFN neurons predicted to be active.

Previous studies have demonstrated that hidden states evolve slowly during inference, exhibiting high similarity between adjacent and even non-adjacent layers (Liu et al., 2023; Zhong et al., 2025). This observation motivates our design of cross-layer sparsity prediction. Specifically, for the sparse predictor ($MLP_{i+k}$) corresponding to the $(i+k)_{th}$ layer, we use the output hidden state $h_i$ of the $i_{th}$ attention layer as input to the predictor. The predictor then outputs a sparsity prediction $z_{i+k} \in R^d$ , where d is the fixed number of neurons in the FFN layer, determined by the model architecture.

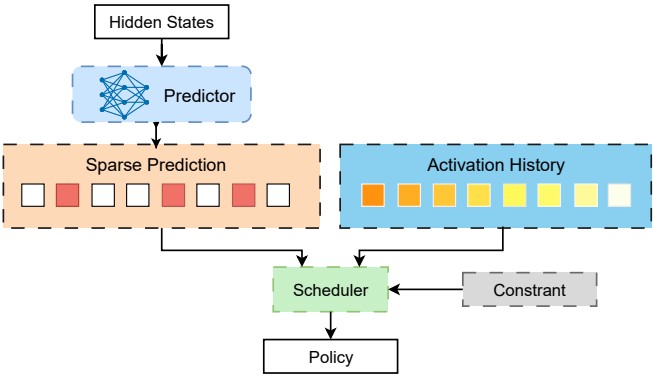

Figure 6: Hierarchical Neural Caching Strategies

### 4.2.2 I/O PIPELINE WITH OVERLAPPED COMPUTATION AND DATA TRANSFER

FFN layers account for the majority of parameters in large language models, and frequent neuron scheduling can result in significant I/O bottlenecks. Even when scheduling is limited to only the neurons predicted to be active, the overhead introduced by data transfer may exceed the performance gains from offloading computation to the CPU. Cross-layer sparsity prediction provides a temporal window between prediction and execution, allowing sufficient time to transfer the required neurons from CPU host memory to GPU memory. Building upon this observation, we propose an I/O pipeline that overlaps computation and transmission.

Specifically, after the attention module of the $i_{th}$ layer completes its computation, a new sub-thread is spawned to perform sparsity prediction and neuron scheduling. While the main thread proceeds with the FFN computation of layer $i$, the sub-thread concurrently predicts the sparsity pattern of layer $i + k$. This thread is synchronized with the CPU-GPU execution of the FFN computation and utilizes CUDA asynchronous transfer mechanisms to prefetch the parameters of the activated neurons in the upcoming layer during ongoing computation. Details regarding data transmission and I/O implementation are provided in Appendix C.3.

### 4.3 HIERARCHICAL NEURAL CACHING STRATEGIES

Prior systems like PowerInfer (Song et al., 2023) statically allocate GPU memory based on offline neuron activation statistics. However, as shown in §3, neuron activations are highly input-dependent and dynamic, making static strategies insufficient for efficient inference. To address this, we propose a hierarchical neural caching mechanism that dynamically adapts neuron placement during runtime based on both sentence-level and token-level activation patterns, thereby improving both resource utilization and inference performance.

### 4.3.1 SCHEDULING METHOD

- **Macro-level Scheduling**: Neuron activation patterns exhibit strong locality when processing the same input sentence. Our system continuously monitors activation frequencies and persistently caches these high-frequency (hot) neurons in GPU memory. This strategy minimizes the overhead from redundant scheduling and data transfers.

- **Micro-level Scheduling**: Even within a single sentence, the set of activated neurons varies significantly across tokens. To address this fine-grained dynamism, we employ an online activation prediction mechanism that leverages the current token's context to estimate the neurons likely to be activated in the next inference step. The system then prioritizes loading these predicted neurons into GPU memory.

### 4.3.2 SCHEDULING STRATEGY MODELING

To implement this hierarchical strategy, we formulate the neuron scheduling task as an optimization problem. The objective is to maximize the total importance of neurons assigned to the GPU, where

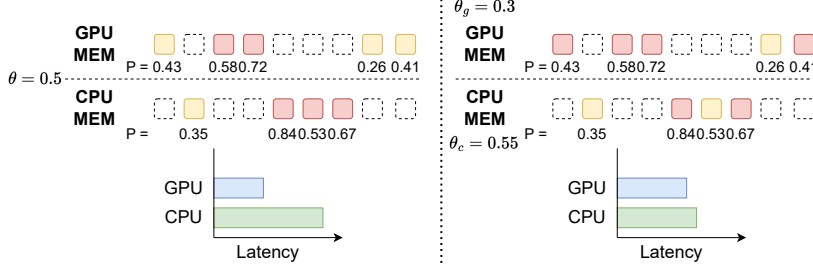

Figure 7: Load-aware Neuron Activation Mechanism

a neuron's importance is jointly determined by its predicted activation for the current step (micro-level) and its historical activation frequency (macro-level).

This optimization is subject to several key constraints:

- **Memory Constraint**: The total memory consumed by neurons allocated to the GPU must not exceed the available memory budget.
- **Communication Constraint**: The transfer of neurons from CPU to GPU must complete within the available computation-I/O overlap window to prevent I/O from becoming a bottleneck.
- **Macro-residency Constraint**: High-frequency neurons are pinned to the GPU to enforce residency and avoid repeated transfers.

Since solving this integer program exactly is computationally prohibitive for real-time inference, we adopt an efficient layer-wise greedy algorithm. This algorithm first guarantees GPU residency for high-frequency neurons, then sorts the remaining neurons by importance, and iteratively loads the highest-ranking ones until the communication constraint is met. This approach strikes a practical balance between overhead and performance. Detailed mathematical modeling for this optimization problem is provided in Appendix E.3.

### 4.4 LOAD-AWARE NEURON ACTIVATION MECHANISM

Existing sparse activation methods often adopt a unified threshold to determine neuron activation (Song et al., 2023; Liu et al., 2023). However, in heterogeneous CPU-GPU systems, this setting is suboptimal due to differing compute capabilities by the heterogeneous computing capabilities.

Our approach dynamically adjusts the threshold based on the workload distribution between CPU and GPU, specifically the number of activated neurons. This dynamic adjustment aims to minimize the overall FFN layer latency, which is dictated by the slower of the two processors figure 7.

To ensure model accuracy is not compromised, we introduce an accuracy-preservation constraint during this adjustment. This constraint guarantees that the prediction error rate does not exceed that of a baseline static threshold. In this way, the system achieves dynamic load balancing between the CPU and GPU, further reducing inference latency without sacrificing accuracy. The detailed modeling and solving method for dynamic threshold adjustment can be found in Appendix E.4.

## 5 EVALUATION

### 5.1 EXPERIMENT SETTING

**Hardware Configuration**. To evaluate the performance and adaptability of DynamicInfer under heterogeneous and consumer-grade hardware environments, we conduct experiments on two different PC configurations:

- **PC-High**: A high-performance workstation featuring an NVIDIA RTX 4090 (24GB) and an Intel Xeon Platinum 8358P CPU. It utilizes a PCIe 4.0 interface (64GB/s bandwidth) and is equipped with 450GB of host memory.

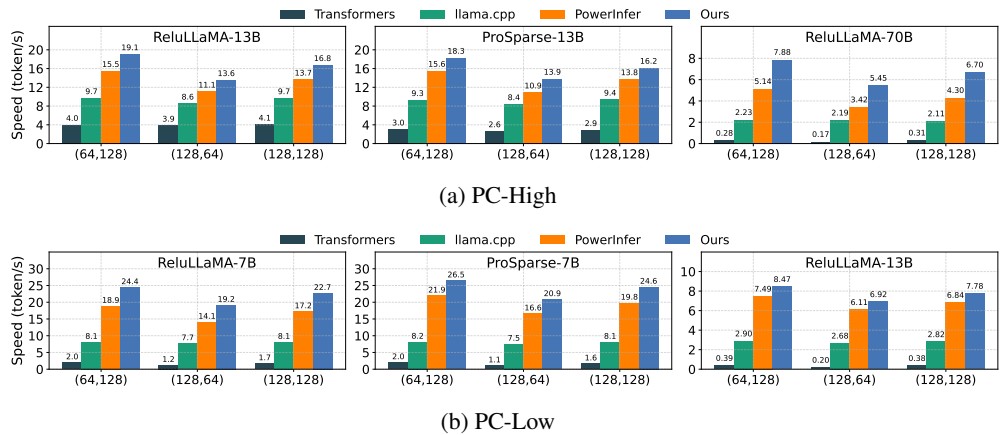

Figure 8: Generation Speed of various models on two hardware. The X axis indicates the input length and output length.

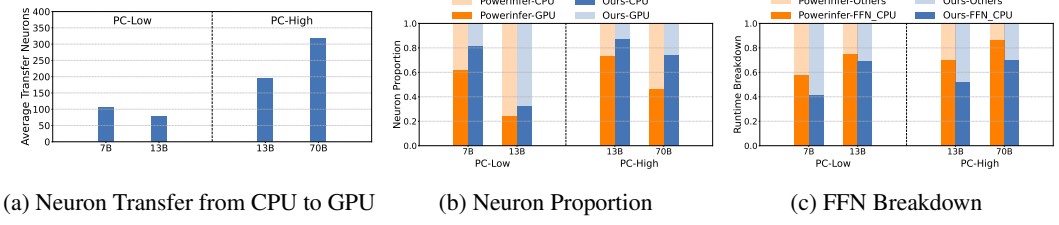

(a) Neuron Transfer from CPU to GPU     (b) Neuron Proportion     (c) FFN Breakdown

Figure 9: Neuron proportion and computation breakdown on different conditions

- **PC-Low**: A standard configuration equipped with an NVIDIA RTX 2080 Ti (12GB) and an AMD Ryzen 9 5950X CPU. It operates on a PCIe 3.0 interface (32GB/s bandwidth) with 120GB of host memory.

**Models**. We evaluate two series of models: ReluLLaMA (Song et al., 2023), Prosparse (Song et al., 2024). On the PC-Low configuration, we test the 7B and 13B model variants. On the PC-High configuration, we test the 13B and 70B variants. Due to memory limitations, the 70B model is quantized to int4, while all other models use FP16 to balance inference speed and accuracy. Intermediate activations are consistently stored in FP32 format.

**Evaluation Metrics**. For load and latency evaluation, we use the Alpaca (Taori et al., 2023) dataset as input. Each latency test is conducted with 100 samples. Unless otherwise specified, we use an input of 64 tokens and generate an output of 128 tokens. The primary metric for latency evaluation is Tokens Per Second (TPS) (NVIDIA), calculated by dividing the total number of tokens generated during the decoding phase by the total response time. This metric provides a quantitative measure of response generation efficiency.

**Baselines**. We compare DynamicInfer against three state-of-the-art baselines:

- **Transformers (Wolf et al., 2020)**: The widely-used framework for large-scale model loading and inference, and integrated with the Accelerate toolkit, supports dynamic offloading to CPU and GPU-centric inference.
- **llama.cpp (Gerganov, 2025)**: An efficient and popular inference engine optimized for local deployment of LLMs.
- **PowerInfer (Song et al., 2023)**: A framework that integrates offloading and sparse activation techniques, achieving latency optimizations in local inference scenarios.

## 5.2 MAIN RESULT

We first measured the end-to-end inference latency of our system in comparison with other systems. Inputs of varying lengths were sampled from the Alpaca dataset to evaluate their TPS performance.

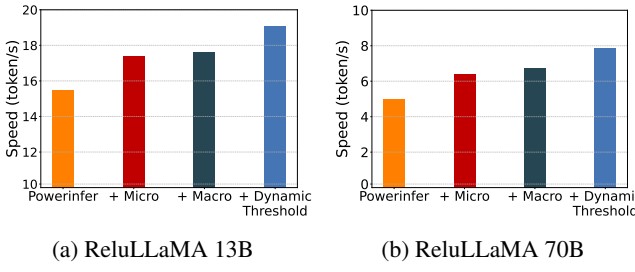

|                          |                          |
| :----------------------: | :----------------------: |
| (a) ReluLLaMA 13B        | (b) ReluLLaMA 70B        |

Figure 10: Ablation Study

We tested three configurations: input lengths of approximately 64 or 128 tokens, and output lengths of 64 and 128 tokens.

On PC-High (figure 8a), DynamicInfer achieves an 18%–26% speedup over PowerInfer and 60%–97% over llama.cpp for the 13B models, and 53%–59% and 148%–253% improvements respectively for the 70B model. The most notable gains are observed with ReluLLaMA-70B, where the longer latency of large models amplifies the benefits of our online scheduling.

On PC-Low (figure 8b), DynamicInfer delivers consistent improvements across smaller models: achieving 20%–35% speedup over PowerInfer and 176%–202% over llama.cpp for 7B models, and 11%–13% and 156%–191% gains respectively for the 13B model. The relatively modest improvement on the 13B model is mainly due to the limited 11GB GPU memory, which restricts neuron placement on the GPU, increases CPU reliance, and reduces scheduling efficiency.

## 5.3 NEURON PROPORTION

During inference, our system dynamically offloads neurons from the CPU to the GPU. figure 9a presents the average number of neurons dynamically transferred per layer under various ReluL-LaMA models and hardware configurations. This mechanism enables our system to activate more neurons.

figure 9b illustrates the proportion of active neurons between the CPU and GPU for the ReluLLaMA model series. The active neuron ratio refers to the proportion of active neurons executed by each processing unit. Notably, on PC-High, DynamicInfer significantly increases the proportion of active neurons handled by the GPU - from an average of 51% to 68%. This indicates that, compared to PowerInfer, DynamicInfer substantially enhances GPU utilization while reducing CPU load.

This result is further corroborated by figure 9c, which shows a reduction in the latency breakdown attributed to CPU computation in the FFN. Under DynamicInfer, the average CPU contribution to feedforward latency decreases from 72% to 57%, highlighting a key factor in the overall speedup achieved by our system.

## 5.4 ACCURACY

Since our approach selectively omits neurons predicted to be inactive, we investigated whether such a method affects the accuracy of LLMs. We conducted experiments on four datasets: RTE (Giampiccolo et al., 2007), PIQA (Bisk et al., 2020), COPA (Roemmele et al., 2011), and Winogrande (Sakaguchi et al., 2021). The result shows that the sparsity method does not affect the accuracy of the model, and the detailed result is in Appendix F.1.

## 5.5 ABLATION STUDY

figure 10 illustrates the contribution of each component to the overall performance acceleration of DynamicInfer. We evaluated both the ReluLLaMA-13B and 70B models on the PC-HIGH platform using a stepwise integration approach, progressively incorporating DynamicInfer features into the PowerInfer baseline.

The Micro-level neuron scheduling yielded the most significant speed improvement, achieving 11.3% and 27.6% acceleration for ReluLLaMA-13B and 70B, respectively. When combined with

Macro-level neuron scheduling, the overall inference speed was further improved by 13.6% and 34.0%. This improvement is primarily attributed to the migration of more neuron computations to the GPU, thereby reducing the computational load of active neurons on the CPU. Finally, by incorporating dynamic thresholding, we achieved an overall inference speedup of 23.1% and 57.1% for the two models.

## 6 CONCLUSION

In this paper, we present DynamicInfer, a novel runtime neuron scheduling and offloading framework for efficient LLM inference on consumer-grade GPUs. DynamicInfer addresses the limitations of static neuron partitioning by introducing an online, input-aware scheduling strategy that dynamically adapts to token-level and sentence-level sparsity patterns. Through a combination of hierarchical neural caching strategies, load-aware neuron activation mechanism, and proactive neuron prefetching, DynamicInfer outperforms llama.cpp and PowerInfer by up to 253.3% and 59.4% respectively in real-world scenarios.

## ACKNOWLEDGMENT

This work is supported in part by the National Key R&D Program of China under grant No. 2024YFC2607400, and the Natural Science Foundation of China under Grant No. 62441233.

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

## A  APPENDIX AND LLM STATEMENT

The contents below are appendix.

In this paper, Large Language Models (LLMs) were employed to enhance the article's quality. Specifically, LLMs were utilized to optimize the contents in the introduction, observation and system design sections. All ideas, data, and conclusions presented in this work were developed solely by the authors, without any involvement of large language models.

## B  RELATED WORK

In this section, we will briefly introduce the related works that motivate DynamicInfer.

### B.1 Sparsity Techniques for LLMs

The computational demands of large language models (LLMs) have spurred research into sparsity on model weights (Han et al., 2016; 2015; Ma et al., 2023; Fang et al., 2023), tokens (Zhang et al., 2023), and activations (Liu et al., 2024a; Zhang et al., 2025), aiming at reducing inference latency and memory usage.

Deja Vu (Liu et al., 2023) introduces a concept of contextual sparsity and predicts it on the fly to speed up the LLM inference with an asynchronous and hardware-aware implementation. CoreInfer (Wang et al., 2024) collects activation statistics during the prefilling stage and selectively activate only a small subset of neurons during decoding. SparseInfer (Shin et al., 2025) offers a training-free predictor for activation sparsity in ReLU-activated LLMs, which also enables efficient inference with negligible accuracy.

### B.2 Offloading Strategies for LLMs' Inference

The large memory requirements of LLMs have led to a growing body of work on offloading techniques that move parts of the model or computation to other system components, such as CPUs, GPUs, and secondary storage, to enable efficient inference under hardware constraints (Gerganov, 2025; Sheng et al., 2023; Ren et al., 2025; Luo et al., 2025).

LLM in a flash(Alizadeh et al., 2024) solves the challenge of effectively operating LLMs that exceed the available capacity of DRAM by storing model parameters in flash memory but bringing them into DRAM on demand. PowerInfer (Song et al., 2023) is a high-speed LLM inference engine designed for consumer-grade GPUs. It exploits the power-law distribution in neuron activation to prioritize the loading of more frequently activated neurons, enabling efficient inference on hardware with limited resources.

### B.3 Optimizations in LLM Serving Systems

As demand for high-throughput LLM applications grows, recent work has also focused on optimizing serving systems through better scheduling (Sun et al., 2024; Agrawal et al., 2024; Zhong et al., 2024), memory management (Fu et al., 2024; Lee et al., 2024b), and model execution strategies (Liu et al., 2024b).

In this field, vLLM (Kwon et al., 2023) is a high-throughput LLM serving engine that employs PagedAttention, a memory management technique that reduces the fragmentation of key-value caches. Orca(Yu et al., 2022) is a distributed serving system that employs iteration-level scheduling and selective batching to improve scalability for transformer-based models.

These work typically focuses on improving throughput under large-batch computation scenarios, and therefore cannot be directly applied to latency-sensitive tasks on local platforms.

## C Implementation Details

### C.1 Code and Development

Our DynamicInfer system is developed based on two major open-source frameworks: PowerInfer (Song et al., 2023) and llama.cpp (Gerganov, 2025). The core inference pipeline is implemented in C++ and CUDA, enabling efficient tensor computation and effective sparsity exploitation. The sparse predictor is implemented in Python.

The sparse predictor is implemented as a lightweight MLP, accounting for less than 10% of the total model parameters. We deploy the sparse predictor on the GPU to fully leverage the GPU's parallel computation capabilities and reduce prediction latency. In our experiments, the inference latency introduced by the predictor accounts for less than 2% of the total inference time.

Table 1: Information of Mobile Devices

| Baseline | Commit Id |
|---|---|
| llama.cpp | e59ea539b83d2c7947c99bd350549364dbba450c |
| PowerInfer | 843195eab85a88a70e6660c5e693b167cce474cf |

## C.2 STORAGE AND PLACEMENT

FFN layers dominate the parameter count in large language models and exhibit significant sparsity. For instance, in ReluLlama 7B and ReluLlama 70B, FFN accounts for 70% of the model parameters. Therefore, during the model loading phase, we prioritize allocating non-FFN components—such as attention layers, output layers, and sparse predictors—to the GPU.

We store the KV cache in CPU memory, thereby freeing up more GPU memory for active neurons. Under small batch sizes, the latency incurred by accessing the KV cache is relatively low and negligible compared to the compute and I/O bottleneck introduced by other components.

Before inference begins, we load the model and determine the initial GPU neuron cache. We employ an Integer Linear Programming (ILP) solver to address this optimization problem. The solver takes the offline neuron activation frequencies as input and, given a GPU memory budget, allocates memory for each FFN module and determines which neurons should be prioritized for loading onto the GPU. The objective is to maximize the expected cache hit rate. This method is similar to the strategy used in PowerInfer (Song et al., 2023).

For the FFN layers, we store the full set of FFN weight parameters on the CPU and preload them into pinned memory to accelerate data transfer from CPU to GPU. On the GPU, we maintain a one-dimensional array, which serves as an index table. During neuron computation, the system utilizes this array to map the input hidden states from the original dimensional space to the submatrix dimensions on the GPU. On the GPU, we maintain a one-dimensional array called $gpu\_bucket$, which serves as an index table. Specifically, $gpu\_bucket[i]$ indicates that the $i_{th}$ row of the GPU submatrix corresponds to the $gpu\_bucket[i]_{th}$ neuron in the original FFN layer. During neuron computation, the system utilizes this array to map the input hidden states from the original dimensional space to the submatrix dimensions on the GPU.

## C.3 TRANSFER AND SYNCHRONIZATION

The data transfer threads run concurrently with the main thread. While the main thread is computing layer L, this thread uses the hidden representations from layer L-1 to perform sparse prediction and neuron scheduling for layer L+k.

The scheduling thread leverages CUDA's asynchronous data transfer mechanisms (e.g., cudaMemcpyAsync) to move data. When a "cold" neuron is predicted to be active, its computation is performed on the CPU, and the results are then transferred back to the GPU. To ensure data consistency, we place a synchronization barrier before the FFN computation of each layer begins. This guarantees that all necessary neuron data transfers and result integrations are complete before the main thread starts processing the next layer, thereby maximizing the hiding of I/O latency while maintaining correctness.

## D EXPERIMENT SETTING DETAILS

### D.1 COMPARISON VERSIONS

The versions of llama.cpp (Gerganov, 2025) and PowerInfer (Song et al., 2023) used in our experiments are listed in Table 1. The open-source version of PowerInfer differs from the implementation described in its original paper. Specifically, the open-source code does not include optimizations for the prefill phase, and its handling of the KV cache also deviates from the paper's description. Therefore, although we did not modify the PowerInfer codebase, the experimental results may differ from those reported in the original publication.

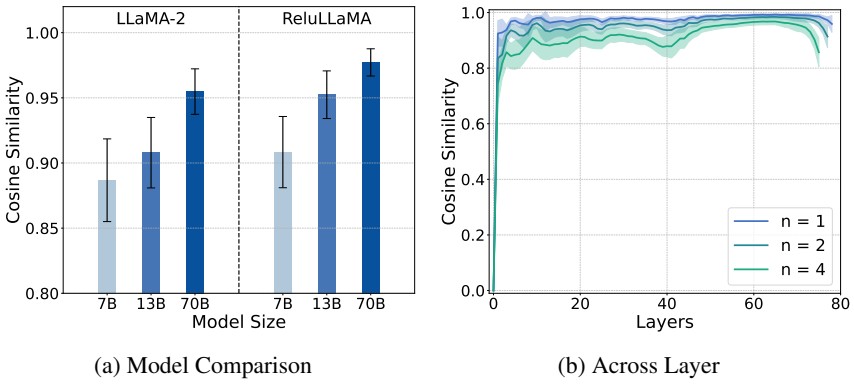

(a) Model Comparison        (b) Across Layer

Figure 11: Hidden State Similarity Across Layers

## D.2 INFERENCE PARAMETERS

All models are evaluated under consistent inference parameters: random seed set to 42, greedy sampling (temperature=0, top_k=1), and a batch size of 1 to simulate real-world interactive scenarios. For llama.cpp, PowerInfer, and our system, the number of threads is set to 8 for CPU-side preprocessing and scheduling, ensuring a fair comparison.

## E METHOD DETAILS

### E.1 SPARSITY PREDICTION DETAILS

In the implementation of PowerInfer, sparse prediction is executed immediately after FFN computation. If we attempt to schedule neurons, it will inevitably introduce significant I/O latency that cannot be ignored.

figure 11a illustrates the cosine similarity of hidden states between adjacent layers across different models, with all models exhibiting an average similarity exceeding 88%. figure 11b presents the cosine similarity between both adjacent and non-adjacent layers in the ReluLLaMA 70B model.

This observation motivates our design of cross-layer sparsity prediction.

### E.2 SINGLE INPUT EXAMPLE

figure 12 illustrates how DynamicInfer operates when processing a specific input, using the prediction of activation states for the next layer as an example. Our system is capable of predicting sparse activations across multiple layers.

Specifically, after the Attention computation of a given layer is completed, the hidden states are forwarded to the sparsity predictor. All neurons are distributed across the GPU and CPU—for example, in the current state, neurons 1, 3, 6, and 7 reside on the GPU, while the remaining neurons are on the CPU. The predictor identifies the neurons that are likely to be activated in future layers; for instance, it may predict that neurons 1, 2, 4, 5, and 8 will be activated. Based on the activation predictions and historical activation probabilities, the scheduler migrates neuron 4 from the CPU to the GPU and evicts neuron 7 from the GPU. The dynamic threshold analysis module then analyzes the current configuration and identifies the CPU as the potential bottleneck for the upcoming computation. As a result, it lowers the activation threshold for the GPU and raises it for the CPU, thereby activating neurons 3 and 6 and suppressing neuron 8.

During this process, the main thread concurrently executes the current layer's FFN computation and the next layer's Attention computation. Once both the main thread's computation and the sub-thread's scheduling are complete, the FFN computation for the next layer begins.

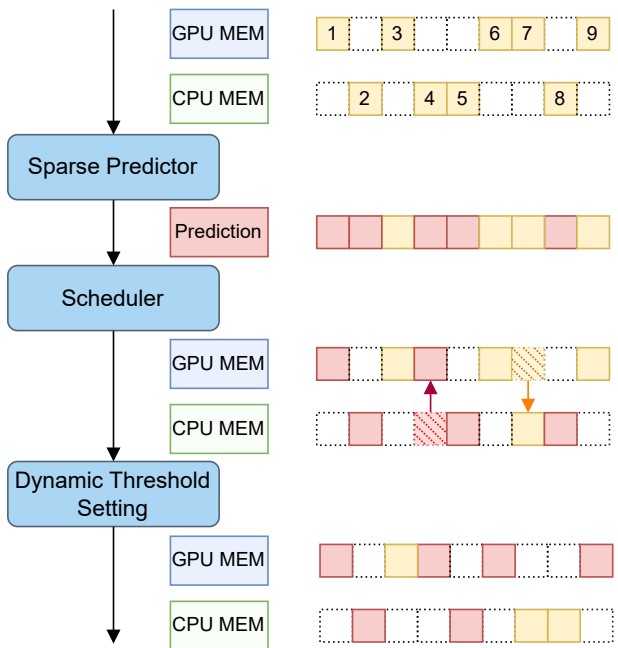

Figure 12: Single Input Example

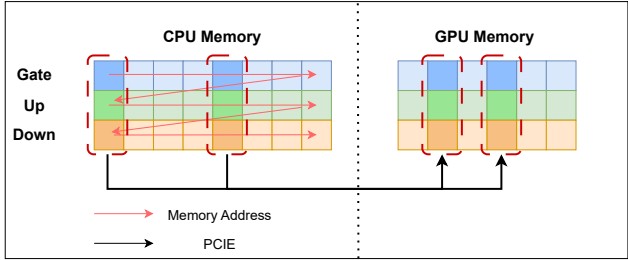

Figure 13: Memory Arrange

Prior to scheduling, the GPU is responsible for computing one neuron and the CPU for four neurons. After scheduling, the GPU computes four neurons while the CPU handles two, thereby achieving dynamic load balancing.

### E.2.1 CONTIGUOUS MEMORY LAYOUT FOR FFN NEURONS

To further optimize the speed of neuron scheduling and data transfer, we redesign the storage layout of FFN neurons. In the LLaMA family of models, the FFN is composed of three submodules: gate, up, and down. Similar upsampling and downsampling modules are also present in other models. Notably, the same set of neurons is activated across all three sublayers.

To streamline this process, we store the gate, up, and down weights of each neuron in a contiguous memory block, reducing the number of reads and improving transfer efficiency. Specifically, on the CPU side, the weights for gate, up, and down layers are stored in a contiguous layout. On the GPU, the sparse matrix representation of these weights is also maintained in a contiguous memory space. During neuron scheduling, we utilize CUDA asynchronous data transfer (e.g., 'cudaMemcpy2DAsync') to move the relevant memory blocks.

### E.3 Scheduling Strategy Details

#### E.3.1 Neuron Importance Metric

The neuron importance metric quantifies the contribution of each neuron, when placed on the GPU, to the overall inference performance of a large language model. We define the importance of neuron $i$ in layer $l$ as:
$$v_{l,i} = a_{l,i} + \lambda f_{l,i},$$
where $a_{l,i} \in \{0, 1\}$ indicates whether the neuron is active at the current inference step (micro-level), and $f_{l,i}$ is its historical activation frequency (macro-level). $\lambda$ is a tunable parameter balancing the two.

#### E.3.2 Optimization Objective

We aim to maximize the total importance of neurons assigned to the GPU:
$$\max \sum_{l,i} g_{l,i} v_{l,i},$$
where $g_{l,i} \in \{0, 1\}$ is a binary variable indicating whether neuron $i$ in layer $l$ is allocated to GPU memory.

#### E.3.3 Constraints

**Memory Constraint:** The placement of neurons is subject to the memory capacity of the processing unit. Specifically, the total memory consumed by neurons allocated to the GPU must not exceed the available GPU memory budget:
$$\sum_{l,i} g_{l,i} M_{single} < M_{budget},$$
where $M_{single}$ denotes the memory footprint of a single neuron and $M_{budget}$ refers to the GPU memory budget available for the FFN layer under the current hardware conditions.

**Communication Constraint:**

The number of neurons loaded onto the GPU is also constrained by intra-layer communication overhead, which is limited by the PCIe bandwidth. To avoid blocking computation, neuron transfers must complete within the available overlap window:
$$t_0 + \sum_{i=0}^{D} g_{l,i} * \left(1 - g'_{l,i}\right) * t_{trans} < \sum_{L=l-k}^{l-1} (t_{other} + t_{FFN,L}),$$
where $g'_{l,i}$ indicates whether neuron $i$ has already transferred to GPU, $t_0$ represents the fixed overhead of the scheduling system, such as function invocations and frequency tracking. $t_{trans}$ denotes the time it takes to transfer one neuron from CPU to GPU. $t_{other}$ is the computation time of non-FFN components in a Transformer layer, and $t_{FFN,L}$ is the FFN computation time of layer $L$ and is modeled as
$$t_{FFN,L} = \max (t_{gpu} * N_{gpu,L}, t_{cpu} * N_{cpu,L} + t_{sync}),$$
where $t_{gpu}$ and $t_{cpu}$ denote the per-neuron computation time on GPU and CPU, respectively. $N_{gpu,L}$ and $N_{cpu,L}$ represent the number of activated neurons in layer $L$ computed on GPU and CPU, respectively. $t_{sync}$ is the synchronization time required to transfer FFN results from CPU to GPU.

In practice, during the Prefill and early Decode stages of inference, we adopt a conservative upper bound on the number of neurons transferred per step and measure the values of $t_0$, $t_{trans}$, $t_{other}$, $t_{gpu}$, $t_{cpu}$, $t_{sync}$. These empirical measurements are then used for solving the optimization objective in subsequent inference steps.

**Macro-level Residency Constraint:** In addition, high-frequency neurons are pinned to the GPU to avoid repeated transfers:
$$h_{l,i} = \begin{cases} 1, & f_{l,i} \geq \phi_{high}, \\ 0, & \text{otherwise}, \end{cases} \qquad g_{l,i} \geq h_{l,i}.$$

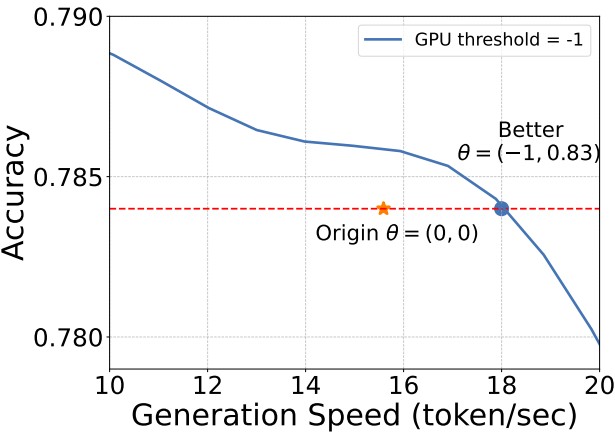

Figure 14: Dynamic Sparsity Profile

where $\phi_{high}$ is a predefined frequency threshold to enforce residency for high-frequency neurons This ensures that neurons identified as high-importance from a Macro-level perspective remain in GPU memory.

### E.3.4    PROBLEM SOLVING

Solving the full integer program is computationally expensive for real-time inference. We adopt a layer-wise greedy strategy using pre-allocated memory budgets $M_{budget,l}$ (from offline ILP in PowerInfer):

$$\sum_i g_{l,i} M_{single} < M_{budget,l},$$

where $M_{budget,l}$ represents the memory allocated to layer $l$, which is determined offline using the ILP solver in PowerInfer based on neuron activation probabilities.

This formulation enables an efficient greedy scheduling strategy that allows selective migration of neurons from CPU to GPU without blocking the main computation thread. In practice, we first identify neurons designated as GPU-resident based on the macro-level residency constraint. Then, we sort the remaining neurons by their importance scores and iteratively load the highest-ranking neurons into GPU memory. Simultaneously, neurons with lower importance scores are evicted, continuing this process until the communication constraint is met.

Moreover, since the per-layer memory budget $M_{budget,l}$ is pre-allocated, there is no need for dynamic GPU memory allocation during each scheduling step. This further reduces runtime overhead and enhances scheduling efficiency.

### E.4    DYNAMIC SPARSITY THRESHOLD DETAILS

Existing sparse activation methods often adopt a unified threshold to determine neuron activation (Song et al., 2023; Liu et al., 2023). However, in heterogeneous CPU-GPU systems, this setting is suboptimal due to differing compute capabilities.

We observe that applying aggressive thresholds on the GPU and conservative ones on the CPU can reduce latency without sacrificing accuracy. For instance, on PIQA (Bisk et al., 2020), setting $\theta_g = -1$ and $\theta_c = 0.83$ achieves comparable accuracy to the unified threshold while reducing latency notably (figure 14).

### E.4.1    MODELING

We treat activation prediction as binary classification. Let $\hat{z}_i = \text{MLP}(x_i)$ be the output of the predictor, trained with cross-entropy loss:

$$L = -(y_i \log \sigma(z_i) + (1 - y_i) \log (1 - \sigma(z_i))),$$

where $\sigma$ denotes the sigmoid function. Assuming the neuron activation follows a Bernoulli distribution, i.e., $y \sim \text{Bernoulli}(\hat{p})$, maximizing the log-likelihood is equivalent to minimizing the cross-entropy loss. Through maximum likelihood estimation, the result $\sigma(\hat{z}_i)$ asymptotically converges to the true activation probability $p_i = P(y = 1 | x_i)$.

A neuron is activated if $p_i > \theta$. The approximation error from skipping neurons under a fixed threshold is:

$$Err = \sum_{p_i < \theta} (1 - p_i) \cdot Err_{single},$$

where $Err_{single}$ is the average error from omitting a single neuron. This formulation captures the trade-off between computational savings and the accuracy degradation introduced by sparsity.

### E.4.2 DYNAMIC THRESHOLD ADJUSTMENT

To balance CPU and GPU workloads, we optimize thresholds to minimize FFN layer latency:

$$\min_{\theta_g, \theta_c} \max \left( t_{gpu} N_{gpu}(\theta_g), \ t_{cpu} N_{cpu}(\theta_c) + t_{sync} \right).$$

where $N_{gpu}(\theta_g)$ and $N_{cpu}(\theta_c)$ denote the number of neurons activated on GPU and CPU under thresholds $\theta_g$ and $\theta_c$, respectively. And because the attention layers are computed on the GPU, CPU-side computations also incur an additional synchronization overhead.

To ensure that inference accuracy is not degraded relative to a baseline static threshold $\theta$, we introduce an accuracy constraint based on the error model:

$$Err(\theta_g, \theta_c) \leq Err(\theta).$$

This constraint can be equivalently expressed as:

$$\sum_{g_j = 1, p_j < \theta_G} (1 - p_j) + \sum_{g_j = 0, p_j < \theta_C} (1 - p_j)$$
$$\leq \sum_{g_j = 1, p_j < \theta} (1 - p_j) + \sum_{g_j = 0, p_j < \theta} (1 - p_j).$$

This is an NP-hard MINLP problem. We adopt a greedy strategy: starting from a static threshold $\theta$, we iteratively decrease $\theta_g$ to shift more computation to the GPU, and increase $\theta_c$ to reduce CPU load, maintaining the error constraint.

The adjustment strategy is GPU load-aware, favoring higher GPU utilization. And the algorithm sacrifices some sparsity to enhance CPU throughput, achieving a more balanced and efficient execution.

### E.5 SYSTEM PARAMETER MEASUREMENT

The constants introduced in our optimization constraints (Section E.3) are determined through a brief offline profiling process. These parameters, including $t_0$, $t_{trans}$, $t_{other}$, $t_{gpu}$, $t_{cpu}$, and $t_{sync}$, are hardware-dependent and reflect the specific system's CPU-GPU I/O bandwidth and computational speeds.

For a given hardware configuration, these values can be treated as constants. The profiling is performed during the initial decoding steps, where we measure the actual timing characteristics of the system. In practice, we adopt conservative estimates for transfer time ($t_{trans}$) to prioritize avoiding computational stalls over maximizing neuron transfer throughput.

We acknowledge that this approach assumes a dedicated execution environment. In scenarios with resource contention from concurrent applications, a more sophisticated online measurement mechanism would be beneficial, which represents a direction for future work.

Table 2: Accuracy

| Model | RTE | PIQA | COPA | Winogrande |
|---|---|---|---|---|
| ReluLLaMA 7B (Dense) | 68.23 | 77.42 | 87.00 | 66.22 |
| ReluLLaMA 7B (Sparse) | 68.59 | 77.47 | 87.00 | 65.98 |
| ReluLLaMA 13B (Dense) | 59.92 | 78.02 | 84.00 | 67.56 |
| ReluLLaMA 13B (Sparse) | 59.57 | 78.07 | 85.00 | 66.93 |
| ReluLLaMA 70B (Dense) | 74.36 | 80.14 | 93.00 | 75.85 |
| ReluLLaMA 70B (Sparse) | 73.28 | 80.41 | 92.00 | 75.06 |
| ProSparse 7B (Dense) | 70.76 | 75.51 | 67.00 | 67.72 |
| ProSparse 7B (Sparse) | 70.40 | 75.13 | 66.00 | 66.53 |
| ProSparse 13B (Dense) | 74.01 | 76.39 | 90.00 | 70.40 |
| ProSparse 13B (Sparse) | 73.64 | 74.10 | 89.00 | 70.33 |

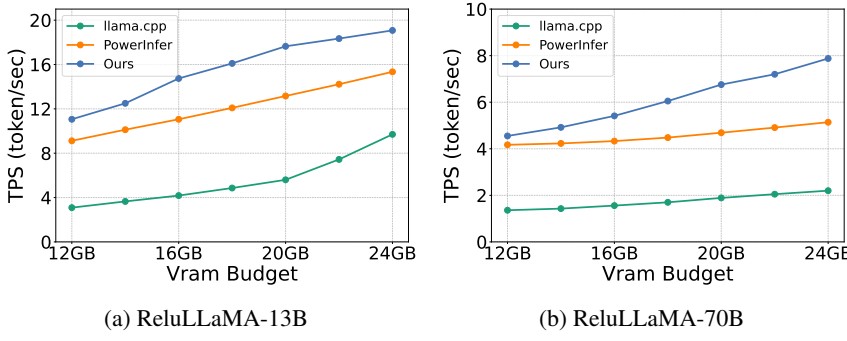

(a) ReluLLaMA-13B  (b) ReluLLaMA-70B

Figure 15: Performance under Different Vram Budget

## F    MORE EXPERIMENT RESULTS

### F.1    ACCURACY EXPERIMENTS DETAILS

We conducted experiments on four datasets: RTE Giampiccolo et al. (2007), PIQA Bisk et al. (2020), COPA Roemmele et al. (2011), and Winogrande Sakaguchi et al. (2021). RTE Giampiccolo et al. (2007) evaluates logical reasoning capabilities in language understanding, PIQA Bisk et al. (2020) tests physical commonsense reasoning, COPA Roemmele et al. (2011) assesses causal reasoning, and Winogrande Sakaguchi et al. (2021) measures commonsense-based language understanding and disambiguation.

Table 2 compares the accuracy of the ReluLLaMA and ProSparse model series. The results demonstrate that across all model sizes and task types, the impact of sparse activation on reasoning capabilities is negligible.

### F.2    PERFORMANCE IN DYNAMIC SCENARIO

We extended our experiments across various scenarios to demonstrate the effectiveness of our approach under different conditions.

figure 15a and figure 15b illustrate the variation in TPS when dynamically managing the GPU VRAM budget on the PC-High configuration. As shown, our method consistently outperforms PowerInfer across a range of GPU VRAM capacities from 12 GB to 24 GB. When the available VRAM decreases, the GPU can accommodate fewer FFN neurons, resulting in relatively smaller performance gains. Conversely, as the VRAM increases, more neurons can be scheduled on the GPU, leading to greater performance improvements.

By setting different initial thresholds, we can obtain varying trade-offs between inference speed and accuracy. Therefore, in systems based on sparse activation prediction, there exists an inherent

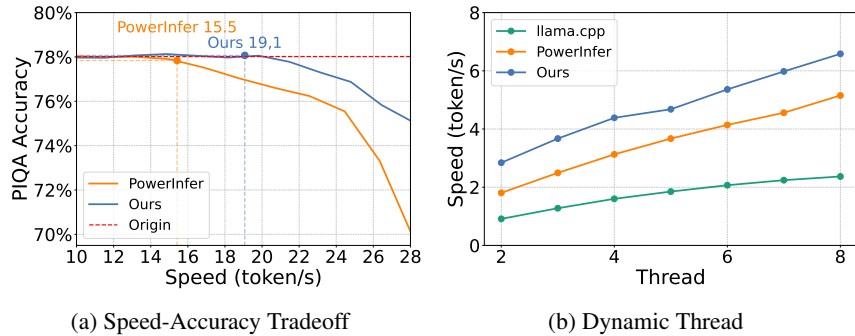

(a) Speed-Accuracy Tradeoff    (b) Dynamic Thread

Figure 16: Speed-Accuracy Tradeoff and Dynamic Thread

Table 3: Peak Memory Usage Comparison (GB)

| Model | Hardware | GPU Memory | | CPU Memory | |
|---|---|---|---|---|---|
| | | PowerInfer | DynamicInfer | PowerInfer | DynamicInfer |
| ReluLlama 7B | 2080Ti | 11.3 | 11.2 | 7.4 | 11.8 |
| ReluLlama 13B | 2080Ti | 11.4 | 11.6 | 21.3 | 36.6 |
| ReluLlama 13B | 4090 | 23.5 | 23.7 | 36.6 | 39.8 |
| ReluLlama 70B (int4) | 4090 | 23.6 | 23.5 | 24.2 | 28.4 |

balance between performance and accuracy. figure 16a illustrates the speed-accuracy curve in the PIQA Bisk et al. (2020) dataset for the ReluLLaMA 13B model running on the PC-HIGH platform under different initial threshold configurations. As observed, DynamicInfer achieves nearly the same accuracy as the dense model while delivering significantly higher inference speed compared to the PowerInfer system.

figure 16b presents the results of the ReluLLaMA-70B model on the PC-high machine under varying thread configurations. Different thread counts correspond to different levels of parallelism and CPU workload. Under various thread configurations, our system achieves a speedup of 166%–211% compared to llama.cpp, and a 27.6%–57.3% improvement over PowerInfer.

### F.3 Memory Usage Analysis

Table 3 details the peak memory usage for both GPU and CPU across different model configurations. DynamicInfer introduces minimal memory overhead compared to PowerInfer.

The additional CPU memory consumption in DynamicInfer is primarily attributed to the sparsity predictor and buffers required for dynamic neuron transfer. The GPU memory footprint remains nearly identical between the two systems, demonstrating that our dynamic scheduling approach does not impose additional GPU memory overhead.

### F.4 Generalization to Non-ReLU Activation Models

While our primary experiments focus on ReLU-activated models that exhibit explicit sparsity, our framework generalizes to models with other activation functions. To demonstrate this, we evaluate DynamicInfer on Llama-2 models, which utilize the SiLU (Sigmoid Linear Unit) activation function.

For non-ReLU activations where exact zero values are rare, we adapt the strategy from CATS Lee et al. (2024a), predicting and pruning neurons with the lowest activation magnitudes. Table 4 presents the results on Llama-2 models.

The results demonstrate that DynamicInfer achieves consistent speedups over baselines on non-ReLU models while maintaining comparable accuracy. This validates that our core contribution—the dynamic neuron scheduling framework—is not limited to ReLU-based models but can be effectively applied to various architectures with appropriate sparsity prediction strategies.

Table 4: Performance on Non-ReLU Models (Llama-2 with SiLU Activation)

| Model | Method | RTE | PIQA | COPA | Winogrande | Speed | Speedup |
|-------|--------|-----|------|------|-----------|-------|---------|
| Llama2-7B (2080Ti) | llama.cpp | 63.90 | 78.07 | 89.00 | 69.38 | 7.9 | 1.00× |
| | PowerInfer | 63.17 | 76.71 | 88.00 | 67.48 | 14.2 | 1.79× |
| | DynamicInfer | 63.54 | 76.55 | 88.00 | 67.56 | 16.5 | 2.09× |
| Llama2-13B (2080Ti) | llama.cpp | 67.15 | 79.43 | 91.00 | 72.45 | 2.5 | 1.00× |
| | PowerInfer | 66.45 | 77.80 | 89.00 | 71.50 | 5.27 | 2.11× |
| | DynamicInfer | 66.01 | 77.69 | 90.00 | 71.43 | 5.81 | 2.32× |
| Llama2-13B (4090) | llama.cpp | 67.15 | 79.43 | 91.00 | 72.45 | 9.8 | 1.00× |
| | PowerInfer | 66.45 | 77.80 | 89.00 | 71.50 | 12.4 | 1.27× |
| | DynamicInfer | 66.01 | 77.69 | 90.00 | 71.43 | 13.8 | 1.41× |

