# OpenReview forum: "DynamicInfer: Runtime-Aware Sparse Offloading for LLMs Inference on a Consumer-Grade GPU"
_ICLR.cc/2026/Conference — ICLR 2026 Poster_

### Official Review · Reviewer_h9mu · 2025-10-18

**Soundness:** 3
**Presentation:** 2
**Contribution:** 2
**Rating:** 4
**Confidence:** 3

**Summary:**

This work proposes a CPU–GPU collaborative inference method for large language models on a consumer-grade GPU by leveraging activation sparsity. Specifically, it introduces a hierarchical neural caching strategy that determines neuron placement based on sentence-level and token-level activation patterns. This differs from the baseline, PowerInfer, which also performs CPU–GPU collaborative inference but relies on offline profiling and a static hot–cold neuron partitioning. Given a layer’s hidden state, the predictor forecasts the sparsity of a single later layer located several layers deeper in the network. To hide host-to-device weight-transfer overhead, the paper applies optimization techniques such as I/O pipelining with overlapping of computation and data transfer. To maintain load balancing between GPU and CPU, it adaptively adjusts the threshold based on the observed loads of each device. The neuron-placement strategy must satisfy memory, communication, and macro-residency constraints; the paper formulates this as an integer linear programming (ILP) problem. Evaluated with ReluLLaMA and Prosparse on the Alpaca dataset, the method achieves a 253% speedup over llama.cpp and a 59% speedup over PowerInfer.

**Strengths:**

- This paper targets an important research problem: democratizing LLM inference and preserving users’ data privacy by enabling inference on a consumer-grade GPU.

- The plots and figures are very helpful for understanding the proposed method, when each component operates, and where the speedup comes from.

- The achieved speedup over the baseline is substantial, and the results show the method works across several model sizes (7B, 13B, 70B).

- Figure 1, which describes the offloading system, is very helpful for understanding the baselines, prior work, and how data transfer occurs in each case.

- The rationale behind the method is sound—both in terms of systems optimization (I/O pipelining, overlapping, and balancing the load) and algorithmic design (dynamic, runtime prediction versus offline, static prediction).

**Weaknesses:**

- The evidence supporting the claim that the method preserves accuracy is insufficient. The paper evaluates only two model architectures (ReluLLaMA and ProSparse). To demonstrate generality, additional experiments on other architectures (e.g., Qwen, DeepSeek) are needed to show that accuracy is maintained.

- The method targets models whose activation functions induce high sparsity by design (e.g., ReLU). The paper should evaluate whether comparable accuracy and speedups hold for architectures using other activations such as ReGLU and SwiGLU.

- The experimental setup needs clearer description and stronger justification. The paper states that a “low-end PC” uses 120 GB of host memory, but 120 GB is arguably not low-end. Clearly specifying the exact memory configuration would improve transparency.

- A sensitivity study is needed for the hyperparameter k and for how speedup varies with input length, output length, and batch size, to establish robustness across diverse settings. Since k controls how far ahead the predictor targets and influences predictor accuracy, its impact should be quantified.

- Stronger baselines are needed. Although “Transformers” is included, it typically shows slower performance than highly optimized inference frameworks (e.g., vLLM) and baselines like SpecInfer that employ speculative decoding.

**Questions:**

- Accuracy/generalization: Beyond ReluLLaMA and ProSparse, can you report accuracy on additional architectures (e.g., Qwen, DeepSeek) to demonstrate that accuracy is preserved more generally?

- Activation functions: Do the proposed gains (accuracy and speedup) hold for models that use non-ReLU activations such as ReGLU or SwiGLU?

- Experimental setup clarity: Could you clarify the “low-end PC” configuration—especially the 120 GB host memory claim—and provide the exact memory specifications to justify this designation?

- Sensitivity/robustness: How sensitive is performance to the hyperparameter k? And how do speedups vary with input length, output length, and batch size?

- Baselines: Can you include stronger baselines (e.g., vLLM, SpecInfer with speculative decoding) and compare throughput/latency directly to these systems?

**Details Of Ethics Concerns:**

This paper does not raise any special ethical concerns.

---

> ### Author Response · Authors · 2025-11-22
> **Author Response to Reviewer h9mu (1/2)**
>
> We sincerely thank the reviewers for their insightful feedback and constructive suggestions, which have been invaluable in strengthening our work. We address the specific concerns below.
>
> ---
>
> ### **Q1: Generality:**
>
> > Do the proposed gains (accuracy and speedup) hold for models that use non-ReLU activations such as ReGLU or SwiGLU?
> > Beyond ReluLLaMA and ProSparse, can you report accuracy on additional architectures (e.g., Qwen, DeepSeek) to demonstrate that accuracy is preserved more generally?
>
> Our method's effectiveness extends to models with non-ReLU activation functions. We conducted experiments using Llama 2 models, which utilize the SiLU (Sigmoid Linear Unit) activation function, a variant of SwiGLU.
>
> For activation functions like SiLU, predicting which neurons will output exact zeros is not feasible. To adapt our approach, we adopted a strategy inspired by CATS [1], where we predict and offload the 50% of neurons with the lowest anticipated activation values. As shown in the table below, this adaptation still yields significant speedups over baselines like PowerInfer and Llama.cpp while maintaining comparable accuracy.
>
> This demonstrates that our core contribution—the dynamic scheduling framework—is not limited to ReLU-based models. It can be effectively applied to various architectures, provided that a suitable method for predicting neuron sparsity is available.
>
> | Model | Method | RTE | PIQA | COPA | Winogrande | Speed (token/s) | Speedup |
> | :--- | :--- | :--- | :--- | :--- | :--- | :--- | :--- |
> | **Llama2-7B + 2080ti** | Llama.cpp | 63.90 | 78.07 | 89.00 | 69.38 | 7.9 | 1.00x |
> | | Powerinfer | 63.17 | 76.71 | 88.00 | 67.48 | 14.2 | 1.79x |
> | | DynamicInfer | 63.54 | 76.55 | 88.00 | 67.56 | 16.5 | 2.09x |
> | **Llama2-13B + 2080ti**| Llama.cpp | 67.15 | 79.43 | 91.00 | 72.45 | 2.5 | 1.00x |
> | | Powerinfer | 66.45 | 77.80 | 89.00 | 71.50 | 5.27 | 2.11x |
> | | DynamicInfer | 66.01 | 77.69 | 90.00 | 71.43 | 5.81 | 2.32x |
> | **Llama2-13B + 4090** | Llama.cpp | 67.15 | 79.43 | 91.00 | 72.45 | 9.8 | 1.00x |
> | | Powerinfer | 66.45 | 77.80 | 89.00 | 71.50 | 12.4 | 1.27x |
> | | DynamicInfer | 66.01 | 77.69 | 90.00 | 71.43 | 13.8 | 1.41x |
>
> <br>
>
> > [1] Lee, Donghyun, et al. "CATS: Context-Aware Thresholding for Sparsity in Large Language Models." *First Conference on Language Modeling.*
>
>
> ---
>
> ###  **Q2: Experimental setup clarity:**
> > Could you clarify the “low-end PC” configuration—especially the 120 GB host memory claim—and provide the exact memory specifications to justify this designation?
>
> We apologize for the confusion. The "120GB" figure referred to the maximum memory capacity supported by the CPU, not the actual host memory used in our experiments. To clarify, our system's performance on a low-end PC is primarily constrained by CPU computational power and PCIe bandwidth, rather than the amount of host memory.
>
> The core requirement is that the combined CPU and GPU memory must be sufficient to hold the model parameters and associated overhead (e.g., KV cache, predictor). As detailed in the table below, DynamicInfer introduces only a minimal memory overhead compared to PowerInfer, primarily for the predictor and dynamic data transfer buffers.
>
> | Model | Hardware | Powerinfer-GPU (GB) | DynamicInfer-GPU (GB) | Powerinfer-CPU (GB) | DynamicInfer-CPU (GB) |
> | :--- | :--- | :--- | :--- | :--- | :--- |
> | ReluLlama 7B | 2080ti | 11.3 | 11.2 | 7.4 | 11.8 |
> | ReluLlama 13B | 2080ti | 11.4 | 11.6 | 21.3 | 36.6 |
> | ReluLlama 13B | 4090 | 23.5 | 23.7 | 36.6 | 39.8 |
> | ReluLlama 70B int4 | 4090 | 23.6 | 23.5 | 24.2 | 28.4 |

---

> ### Author Response · Authors · 2025-11-22
> **Author Response to Reviewer h9mu (2/2)**
>
> ###  **Q3: Sensitivity/robustness:**
>
> > How sensitive is performance to the hyperparameter k? And how do speedups vary with input length, output length, and batch size?
>
> ***Sensitivity to Hyperparameter k***
>
> The hyperparameter `k` determines the number of future layers the scheduler considers for neuron migration. A larger `k` provides a longer time window for data transfer but introduces two trade-offs:
> 1.  **Predictor Accuracy:** Predicting neuron activity further into the future is inherently less accurate, which can lead to a decline in overall model accuracy.
> 2.  **I/O Contention:** Migrating neurons for multiple layers simultaneously can create contention for PCIe bandwidth. This makes the transfer time (`t_trans`) variable and difficult to model as a constant, potentially leading to pipeline stalls and negating performance gains.
>
> Our experiments with ReluLlama 7B show that performance peaks at `k=2`. Beyond this, the negative effects of reduced predictor recall and I/O contention begin to outweigh the benefits of a longer migration window. Therefore, we set `k=2` for all experiments.
>
> | Model | k | Predict Recall | RTE | PIQA | COPA | Winogrande | Speed (token/s) |
> | :--- | :-: | :--- | :--- | :--- | :--- | :--- | :--- |
> | **ReluLlama 7B** | 0 | 0.96 | 68.23 | 77.42 | 87.00 | 66.21 | 18.9 |
> | | 1 | 0.94 | 68.47 | 77.58 | 86.00 | 66.16 | 21.7 |
> | | **2** | **0.93** | **68.47** | **77.47** | **87.00** | **65.96** | **23.5** |
> | | 3 | 0.88 | 67.51 | 76.95 | 85.00 | 65.28 | 22.6 |
> | | 4 | 0.84 | 66.43 | 76.71 | 84.00 | 64.36 | 20.8 |
>
> ***Impact of Input/Output Length and Batch Size***
>
> *   **Batch Size:** Our method is optimized for personal and edge-side inference, where `batch_size=1` is the most common scenario. As batch size increases, neuron activation patterns become denser (a neuron is computed if activated by *any* input in the batch), which reduces the sparsity our system can exploit. As shown below, DynamicInfer maintains a performance advantage for smaller batch sizes, but this advantage diminishes as the batch size grows.
>
> | Model | Hardware | Batch Size | Llama.cpp (ms/token) | DynamicInfer (ms/token) | Speedup |
> | :--- | :--- | :--- | :--- | :--- | :--- |
> | **ReluLlama 13B** | 2080Ti | 1 | 2.9 | 8.47 | 2.92x |
> | | | 2 | 5.4 | 11.86 | 2.19x |
> | | | 4 | 9.28 | 13.55 | 1.46x |
> | | | 8 | 13.92 | 17.62 | 1.27x |
>
> *   **Input Length:** Longer input sequences lead to a denser prefill phase, as more neurons are activated to process the context. For very long inputs, we recommend using a dense computation method for the prefill phase and activating DynamicInfer only for the token generation (decode) phase. This hybrid strategy is also employed by our baseline, PowerInfer.
>
> *   **Output Length:** Our speedup is most pronounced during the decode phase, where significant sparsity can be exploited for each generated token. While the Time Per Output Token (TPOT) remains relatively constant, the overall throughput in Tokens Per Second (TPS) increases with longer output sequences because the accelerated decode phase constitutes a larger portion of the total execution time.
>
> ---
>
> ### **Q4: Baselines:**
> > Can you include stronger baselines (e.g., vLLM, SpecInfer) and compare throughput/latency directly to these systems?
>
> **Comparison with vLLM**
>
> We compared DynamicInfer with vLLM using its `cpu_offload` feature. This feature operates similarly to a standard transformer implementation, swapping entire layers between CPU and GPU. This approach incurs substantial data transfer latency, making it inefficient for the low-batch, resource-constrained scenarios our work targets. As the results show, DynamicInfer significantly outperforms vLLM with CPU offloading.
>
> | Model | Hardware | **(in=64, out=128)** | **(in=128, out=64)** | **(in=128, out=128)** |
> | :--- | :--- | :---: | :---: | :---: |
> | **ReluLlama-7B** | 2080ti | 2.1 | 1.4 | 1.8 |
> | **ReluLlama-13B** | 2080ti | 0.5 | 0.36 | 0.44 |
> | **ReluLlama-13B** | 4090 | 3.8 | 3.6 | 4.2 |
> | **ReluLlama-70B (int4)** | 4090 | 0.4 | 0.25 | 0.35 |
>
> **Orthogonality to Speculative Decoding (e.g., SpecInfer)**
>
> Our method, DynamicInfer, is orthogonal to and complementary with speculative decoding techniques. The two approaches address different bottlenecks:
> *   **DynamicInfer** accelerates the computation of a *single token* by exploiting dynamic activation sparsity.
> *   **Speculative Decoding** accelerates generation by producing *multiple tokens* in one step, using a smaller model to predict the output of a larger one.
>
> DynamicInfer can be integrated with speculative decoding. Since our system supports batch sizes greater than 1, it can readily accommodate the verification of draft tokens generated by a smaller model. This combination would allow the verification step of speculative decoding to also benefit from sparsity-based acceleration. Integrating these two approaches is a promising direction for future work.

---

### Official Review · Reviewer_N69j · 2025-10-26

**Soundness:** 2
**Presentation:** 3
**Contribution:** 2
**Rating:** 4
**Confidence:** 4

**Summary:**

DynamicInfer improves LLM inference throughput served with memory-constrained GPUs via sparsification. It does this by sparsifying the neurons in the Feed Forward Network (FFN) stages of transformers. The key contribution is a scheme for 'online sparsification', where the sparsity map of the next layer is estimated with the previous layer activations, giving enough time for the CPU-GPU memory transfers of the FFN weights to happen. DynamicInfer Is tested with LLM architecture that utilize ReLU activations and does not markedly drop LLM accuracy on four datasets.

**Strengths:**

Thank you for submitting your work. I overall like the paper. In terms of strengths:
* I like the idea of online sparsity. It's a complicated technique, but there are few alternatives when memory restrictions are aggressive (except for quantization, which should be compared to).
* The paper is mostly well-presented. §2 lays the groundwork for the rest of the discussion, and §3 conveys the general idea (though I have suggestions for improving §3).
* DynamicInfer is evaluated on two machines, 5 models, and compared to 3 other schemes.

**Weaknesses:**

I find 3 major flaws in the paper:
* From what I know, most modern LLMs do not use ReLU activation functions. Older LLMs used GeLU. The previous generation, such as Llama3, Qwen2 used SiLU, and the newest generation such as gpt-oss use SwiGLU. These families of activations always contribute to the next layer neurons, regardless of their value. ReLU, on the other hand, is akin to a on/off switch; some neurons do not affect the next layer at all, which is why sparsification would be a prime optimization candidate. This makes me worry if DynamicInfer would incur more substantial accuracy losses with commonly used LLMs, such as the original Llama family.
* The problem setup is a memory constrained GPU, and model compression is allowed (sparsification that skips neurons is technically model compression). So a natural contender is quantization. The PC-low setup with 7B models and PC-high setup with 13B models can be compared to int8 quantized versions (done properly, as simply casting would break the model) of the original models. I understand that quantization cannot work with 70B models, but they are still simpler alternatives than sparsification.
* The input/output sequence lengths are too small compared to real workloads. Even the relatively old ShareGPT dataset has input/output sequence lengths in the 300/200 range, and more modern datasets have even higher input lengths (due to longer prompts) and output lengths (due to reasoning chains). I wonder if the benefits of sparsification still hold at longer sequence lengths, since the attention runtime will increase and become a bigger share of the full inference pass.

Some minor comments:
* I found the explanation of the figures in §3 to be confusing. For example, why are there two bars in every group in Figures 2b and 2c? I expected a single bar to show the Jaccard distance between global hot neurons and per sentence hot neurons (Also, please cite a reference for Jaccard distance, or use the more common IoU metric which is quite similar). As another example, the oracle is defined to be $\frac{\text{\\# activated neurons}}{\text{\\# GPU neurons}}$. If this is the definition, then what is "Oracle-GPU" and "Oracle-CPU", and why do they sum to 1?
* The paper has formatting issues (figures have inconsistent reference formatting) and needs proofreading (typos).

**Questions:**

I am willing to increase my score if the authors can either provide reasonable answers to, or provide empirical evidence for the following questions:
* Can DynamicInfer improve the throughput of modern LLMs, e.g., Llama3 family or gpt-oss?
* How does DynamicInfer compare against vLLM with quantized models (throughput and accuracy)?
* How do DynamicInfer's improvements change at ISL/OSL 2K/128?

---

> ### Author Response · Authors · 2025-11-22
> **Author Response to Reviewer N69j (1/2)**
>
> We sincerely thank the reviewers for their insightful feedback and constructive suggestions, which have been invaluable in strengthening our work. We address the specific concerns below.
>
> ---
>
> ### **Q1: Performance with Non-ReLU Models and Modern LLMs**
> > Can DynamicInfer improve the throughput of modern LLMs, e.g., Llama3 family or gpt-oss?
>
> Thank you for this important question. Our framework is not limited to ReLU-based models. To demonstrate this, we conducted experiments on Llama2 models, which utilize the SiLU activation function. The results are presented below.
>
> | Model                 | Method       | RTE   | PIQA  | COPA  | Winogrande | Speed (tokens/s) | Speedup |
> | --------------------- | ------------ | ----- | ----- | ----- | ---------- | ---------------- | ------- |
> | **Llama2-7B + 2080Ti**  | Llama.cpp    | 63.90 | 78.07 | 89.00 | 69.38      | 7.9              | 1.00x   |
> |                       | PowerInfer   | 63.17 | 76.71 | 88.00 | 67.48      | 14.2             | 1.79x   |
> |                       | DynamicInfer | 63.54 | 76.55 | 88.00 | 67.56      | 16.5             | 2.09x   |
> | **Llama2-13B + 2080Ti** | Llama.cpp    | 67.15 | 79.43 | 91.00 | 72.45      | 2.5              | 1.00x   |
> |                       | PowerInfer   | 66.45 | 77.80 | 89.00 | 71.50      | 5.27             | 2.11x   |
> |                       | DynamicInfer | 66.01 | 77.69 | 90.00 | 71.43      | 5.81             | 2.32x   |
> | **Llama2-13B + 4090**   | Llama.cpp    | 67.15 | 79.43 | 91.00 | 72.45      | 9.8              | 1.00x   |
> |                       | PowerInfer   | 66.45 | 77.80 | 89.00 | 71.50      | 12.4             | 1.27x   |
> |                       | DynamicInfer | 66.01 | 77.69 | 90.00 | 71.43      | 13.8             | 1.41x   |
>
> While non-ReLU activations like SiLU do not produce exact zero values, they exhibit "near-sparse" behavior where many neurons have minimal activation values. We can predict these low-activation neurons, a technique validated by prior work (e.g., CATS[1]).
>
> Crucially, the core contribution of our paper is the **dynamic neuron scheduling framework**, which is independent of the specific sparsity prediction strategy. As long as a reasonable sparsity prediction method can be implemented for a given activation function (including those in Llama3 or other modern LLMs), our dynamic scheduling mechanism can be applied to accelerate inference.
>
> > [1] Lee, Donghyun, et al. "CATS: Context-Aware Thresholding for Sparsity in Large Language Models." *First Conference on Language Modeling.*
>
> ---
>
> ### **Q2: Comparison with Quantization Methods**
> > How does DynamicInfer compare against vLLM with quantized models (throughput and accuracy)?
>
> We would like to clarify a critical point: **DynamicInfer is orthogonal and complementary to quantization**, not a mutually exclusive alternative.
>
> *   **Quantization** addresses **storage density**. It reduces the model's memory footprint by lowering the bit-width of parameters (e.g., from FP16 to INT4), enabling larger models to fit into limited GPU memory.
> *   **DynamicInfer** addresses **computational and I/O efficiency**. It reduces the runtime computational load by predicting and offloading only the "active" neurons for computation, optimizing the use of heterogeneous hardware (CPU/GPU).
>
> These two techniques can be combined for a synergistic effect. In fact, we have already demonstrated this in our paper. As stated in Section 5.1 ("Models"):
> > *"Due to memory limitations, the 70B model is quantized to int4, while all other models use FP16..."*
>
> This shows that for the most demanding 70B model, we first applied INT4 quantization and then leveraged our DynamicInfer framework. Even after quantization, an INT4 70B model (~35-40GB) still exceeds the memory of a consumer GPU like the RTX 4090 (24GB), making offloading necessary. This highlights the value of our work: when quantization alone is insufficient to fit a model onto the GPU, an efficient runtime offloading system like DynamicInfer becomes essential for achieving high performance.
>
> In summary, quantization does not eliminate the need for offloading. Our approach provides critical acceleration in this "post-quantization" scenario, where model size still surpasses available GPU memory.

---

> ### Author Response · Authors · 2025-11-22
> **Author Response to Reviewer N69j (2/2)**
>
> ### **Q3: Impact of Different Input/Output Lengths**
> > How do DynamicInfer's improvements change at ISL/OSL 2K/128?
>
> Thank you for the question. We evaluated our method with a long input sequence length (ISL=2K) and an output sequence length (OSL) of 128. The results are as follows:
>
> | Model           | Method       | Speed (tokens/s) | Speedup |
> | --------------- | ------------ | ---------------- | ------- |
> | **ReluLlama-13B** | Llama.cpp    | 9.21             | 1.00x   |
> |                 | PowerInfer   | 11.33            | 1.23x   |
> |                 | DynamicInfer | 12.31            | 1.34x   |
> | **ReluLlama-70B** | Llama.cpp    | 2.05             | 1.00x   |
> |                 | PowerInfer   | 3.02             | 1.47x   |
> |                 | DynamicInfer | 3.72             | 1.81x   |
>
> During the prefill phase with very long input sequences, a larger proportion of neurons are activated to process the context. In this scenario, the benefits of sparse computation are outweighed by the overhead of the sparsity predictor. The key consideration here is that as the prefill length increases, sparsity tends to decrease since each computation needs to account for the activation requirements of all tokens. In contrast, decoding maintains high sparsity as it processes only one token at a time.
> Therefore, for the prefill phase with long inputs, we adopt a dense computation mode. This strategy is also noted in the implementation of our baseline, PowerInfer. Our speedup is primarily achieved during the token generation (decoding) phase that follows prefill.
>
> ---
>
> ###  **Q4: Other Clarifications**
>
> *   **Writing and Citations:** We appreciate the suggestions regarding the manuscript's presentation. We will carefully revise the paper to improve clarity, correct citations, and ensure consistency between text and figures.
> *   **Explanation of Figures 2b & 2c:** The bar labeled `* between global` represents the Jaccard distance between dynamically observed active neurons and those predicted by a static, offline method (akin to PowerInfer). The significant distance shown in this bar is intended to illustrate that offline, input-agnostic prediction of active neurons is insufficient and fails to capture the dynamic nature of activation sparsity.
> *   **Definition of Oracle:** The "Oracle" in our experiments represents a theoretical performance upper bound. It is defined under the assumption of infinite CPU-GPU bandwidth, where data transfer is instantaneous. In this ideal scenario, all neurons required for computation can be placed on the GPU without any I/O cost. By showing the performance gap between PowerInfer and this Oracle, we demonstrate that GPU computational resources are underutilized. Our method, DynamicInfer, is designed to bridge this gap by intelligently managing computation and data movement under real-world, finite bandwidth constraints, thereby moving closer to the Oracle's performance.

---

### Official Review · Reviewer_CLC1 · 2025-10-30

**Soundness:** 3
**Presentation:** 3
**Contribution:** 3
**Rating:** 6
**Confidence:** 4

**Summary:**

The paper proposes DynamicInfer, a runtime-aware system for efficient LLM inference on resource-constrained devices. It enhances efficiency by dynamically and finely scheduling FFN neurons between CPU and GPU. Its core mechanisms involve using a lightweight sparse predictor to forecast activation patterns online and dynamically adjusting the sparsity threshold based on the overlapping window of computation and data transfer, thereby optimizing hardware utilization. Experiments show that this method significantly reduces end-to-end latency compared to baseline approaches.

**Strengths:**

1. The work addresses the critical challenge of deploying massive LLMs under tight memory constraints by combining model offloading with sparse activation. Unlike prior systems such as PowerInfer that rely on static neuron partitioning based on offline profiling, DynamicInfer introduces a dynamic, input-aware scheduling mechanism that adapts neuron placement between CPU and GPU at runtime. The method dynamically adjust the activated neuron with optimization from system level to minimize the latency. Practical and Timely Problem: The focus on enabling high-performance LLM inference on consumer-grade hardware is highly relevant and impactful. With growing interest in on-device AI, this work provides a compelling solution to a real-world deployment bottleneck.
2. Strong Technical Innovation: The shift from static to runtime-adaptive scheduling is a key conceptual advance. The design of cross-layer sparsity prediction, combined with a greedy yet effective optimization framework, enables fine-grained, input-dependent control over neuron placement—a clear improvement over fixed partitioning schemes.
3. Well-Rounded System Design: The integration of three synergistic components—dynamic caching, load balancing via threshold tuning, and I/O pipelining—forms a cohesive and robust system. The attention to engineering details (e.g., contiguous memory layout, asynchronous transfers) enhances practicality and performance. The method constrains the memory, communication and macro residency to balance dynamic adaptability

**Weaknesses:**

1. As far as I know, the I/O similarity between the initial layers and deep layers of LLMs is quite low—in some other models, this phenomenon may be even more pronounced. In such cases, does cross-layer sparsity prediction still work effectively for these layers or hidden states?
2. Moreover, do you have any additional mechanisms or fallback strategies specifically designed for those exceptional layers with very low similarity?
3. The data transfer between CPU and GPU is rather slow compared with GPU memory access. In your article, does the cross layer prediction fully cover the time of weight transfer? It'll be helpful if you provide detailed data.
4. There are some typos and mismatch of your illustrations with your pictures that may cause misunderstanding. The "Figure3 B" in Observation one seems to be Figure2 B?

**Questions:**

1. Does cross-layer sparsity prediction still work effectively for layers that have low I/O similarity?
2. Moreover, do you have any additional mechanisms or fallback strategies specifically designed for those exceptional layers with very low similarity? Besides, can you provide some data to show cross-layer prediction has mere impact on prediction accuracy?
3. The data transfer between CPU and GPU is rather slow compared with GPU memory access. In your article, does the cross layer prediction fully cover the time of weight transfer? It'll be helpful if you provide detailed data.

---

> ### Author Response · Authors · 2025-11-22
> **Author Response to Reviewer CLC1 (1/2)**
>
> We sincerely thank the reviewers for their insightful feedback and constructive suggestions, which have been invaluable in strengthening our work. We address the specific concerns below.
>
> ---
>
>
> ### **Q1: Effectiveness of Cross-Layer Sparsity Prediction on Layers with Low Similarity**
>
> > Does cross-layer sparsity prediction still work effectively for layers that have low I/O similarity?
>
> Thank you for this insightful question. Our cross-layer prediction method remains effective even for deeper layers where I/O similarity may be comparatively lower. We present the results for ReluLLaMA-7B with a prediction distance of k=2 layers.
>
> | Model | Input Layer | Predict Layer | Recall | F-score |
> | :--- | :--- | :--- | :--- | :--- |
> | ReluLLaMA-7B | 0 | 2 | 0.95 | 0.86 |
> | ReluLLaMA-7B | 8 | 10 | 0.94 | 0.83 |
> | ReluLLaMA-7B | 16 | 18 | 0.92 | 0.79 |
> | ReluLLaMA-7B | 24 | 26 | 0.91 | 0.80 |
> | ReluLLaMA-7B | 29 | 31 | 0.91 | 0.82 |
>
> As shown in the table, the predictor maintains high recall and strong F1-scores in deeper layers. We attribute this robustness to two main factors:
>
> 1.  **Pattern Learning over Similarity:** Our MLP-based predictor is designed to learn complex relationships between layers. The existence of a learnable pattern, rather than high direct similarity, is the primary condition for accurate prediction.
> 2.  **Adaptation to Distribution Shifts:** The predictor implicitly learns to account for varying activation distributions across the model. Deeper layers often exhibit different activation ratios and patterns compared to shallower ones, and our model adapts to these shifts.
>
> Therefore, we are confident that our approach maintains high prediction accuracy across the model's depth.
>
>
> ---
>
> ### **Q2: Fallback Strategies and Impact on Accuracy**
>
> > Moreover, do you have any additional mechanisms or fallback strategies specifically designed for those exceptional layers with very low similarity? Besides, can you provide some data to show cross-layer prediction has mere impact on prediction accuracy?
>
> In our current implementation, we do not employ an explicit fallback mechanism. Our design choice is supported by two key observations:
>
> 1.  **Effectiveness without Fallback:** Our experiments, along with related work such as DejaVu [1], demonstrate that cross-layer predictors can achieve high accuracy without fallback strategies. As shown in our response to Q1, prediction remains accurate even in deeper layers. This suggests that high layer-to-layer similarity is a sufficient but not necessary condition for effective prediction. The crucial factor is the existence of a relationship that our MLP predictor can learn.
>
> 2.  **Impact on Model Accuracy:** The goal of sparsity prediction is to accelerate inference without altering the computational graph or the final output. Our method only predicts which neurons will be zero and prunes them from the computation for a given input. The underlying model weights and the non-pruned computations remain identical. Therefore, if the sparsity prediction is highly accurate (i.e., we only prune true zeros), the impact on the model's output accuracy is mathematically low. The high recall and F1-scores we achieve indicate that prediction errors are minimal, leading to a negligible impact on the final model output.
>
> Nevertheless, we thank the reviewer for this valuable suggestion. A fallback strategy could potentially enhance robustness for outlier cases, and we plan to explore its benefits in future work.
>
> > [1] Liu, Zichang, et al. "Deja vu: Contextual sparsity for efficient llms at inference time." International Conference on Machine Learning. PMLR, 2023.

---

> ### Author Response · Authors · 2025-11-22
> **Author Response to Reviewer CLC1 (2/2)**
>
> ### **Q3: Overlapping Weight Transfer Time**
>
> > The data transfer between CPU and GPU is rather slow compared with GPU memory access. In your article, does the cross layer prediction fully cover the time of weight transfer? It'll be helpful if you provide detailed data.
>
> Thank you for raising this important point. The weight transmission time is managed by the Communication Constraint detailed in Appendix E of our paper.
>
> The constants in this constraint are hardware-dependent and are profiled offline. This is a practical approach as they remain stable on a given hardware configuration. In our implementation, we deliberately set the transmission time constant (`t_trans`) to a conservative (i.e., larger) value. This strategy prioritizes avoiding computational stalls, which incur a significant performance penalty, over transmitting the maximum possible number of neurons.
>
> While a perfect overlap between transmission and computation is difficult to achieve in practice, our method is effective at hiding most of the latency. Our empirical evaluation on an RTX 4090 GPU shows that the ratio of transmission time to computation time (i.e., the ratio of the left to the right side of our Communication Constraint) falls within the **46% to 113%** range in 80% of cases. This indicates that we effectively utilize the computation window to hide the data transfer latency, achieving our goal of overlapping communication and computation.
>
> ---
>
> ### **Q4: Other Problems**
>
> We thank the reviewer for their careful reading and for identifying the error regarding Figure 3(b). The reviewer is correct; this was a typographical error in the manuscript. We have corrected this and will perform a thorough review to ensure all figures and their corresponding descriptions are consistent.

---

### Official Review · Reviewer_uAnA · 2025-10-31

**Soundness:** 2
**Presentation:** 2
**Contribution:** 3
**Rating:** 4
**Confidence:** 4

**Summary:**

This paper presents DynamicInfer, a runtime neuron offloading framework for efficient LLM inference on consumer-grade GPUs. The key innovation is dynamic neuron scheduling that adapts to input-dependent activation patterns, in contrast to PowerInfer's static offline partitioning. The system achieves significant speedups (up to 253% over llama.cpp and 59% over PowerInfer) through three main mechanisms: hierarchical neural caching, load-aware neuron activation, and activation-aware prefetching.

**Strengths:**

**Well motivated Problem**
- Figure 2,3 underlines the importance of dynamic neuron activations.
- The two observations clearly clarify the drawbacks of static partitioning that PowerInfer uses.

**Comprehensive Engineering efforts**
- Authors implement the core inference engine in C++ and CUDA for efficiency. The three component approach is well integrated to minimize overhead between various parts.
- Constraint design shows a thorough understanding of latency profiling.
- Contiguous memory layout (Appendix E.2.1) for FFN neurons is interesting.
- Benchmarked on two different generations of NVIDIA GPUs with different memory bandwidths.

**Weaknesses:**

**Need Additional Comparisons.**
-   As authors note, this is not the first work to dynamically detect activations across the layers. SparseInfer (Shin et al., 2025) (cited by authors) also provides training-free prediction for ReLU-activated LLMs. A direct comparison is essential since both papers target the same models and claim similar benefits. At minimum, discuss architectural differences and when each approach is preferable.
-   Section D.1 states that the open source implementation of PowerInfer differs from the main paper. The results are hence unfair to compare. Authors should ideally contact the authors to implement and/or discuss the implications to the results.
-   Latency evaluation on at least one more dataset can help show robustness.

**Insufficient details on the proposed approach.**
-   How accurate is the trained sparse predictor for new samples? across layers? across models?
-   A number of constants are introduced w.r.t the optimization problem and constraints in E.3. How are they measured? How much do they depend on the hardware?
-   E.4.1 is overly simplified. Lacks justification for Bernoulli distribution. “average error” can vary across layers and models. How is k decided for the sparsity predictor beforehand? Is it data/hardware/model agnostic? what is the procedure?
-   Can you explain the difference between your Macro-level residency constraint and PowerInfer? Although it makes sense to introduce, it does not seem very important from Fig10.
-   Line 1048 abruptly switches from the optimization problem to the greedy strategy. Can you discuss the differences in optimal solution and how much it affects? At the very least, an empirical comparison on some instances would be helpful.
-   Can you provide a table showing peak memory usage for GPU and CPU for PowerInfer and DynamicInfer?

**Writing needs polish.**
-   The references are not clickable.
-   The visual and text layer of the document are mismatched, making it hard to select text.
-   The majority of details about the method are in the appendix.
-   Captions should contain more details.
-   A Grammar check would be helpful.

**Questions:**

-   While the comparison with PowerInfer is reasonable, can you justify comparing your method against llama.cpp? It supports a much larger class of models (non-ReLU, quantization, etc) and was not tailored for efficient inference. It seems unfair.
-   How sensitive is the method w.r.t to λ, the neuron importance metric.
-   Line 361, why did you mention “SparseInfer” ?

---

> ### Author Response · Authors · 2025-11-22
> **Author Response to Reviewer uAnA (1/4)**
>
> We sincerely thank the reviewers for their insightful feedback and constructive suggestions, which have been invaluable in strengthening our work. We address the specific concerns below.
>
> ---
>
> ### **Weakness 1.1: Comparison with Additional Baseline (SparseInfer)**
>
> > As authors note, this is not the first work to dynamically detect activations across the layers. SparseInfer also provides training-free prediction for ReLU-activated LLMs. A direct comparison is essential. At minimum, discuss architectural differences and when each approach is preferable.
>
> Thank you for this suggestion. The core contribution of DynamicInfer is a CPU-GPU scheduling framework that leverages the *dynamic nature* of sparsity. Our framework is agnostic to the specific sparsity predictor used.
>
> To provide a direct and fair comparison, we integrated SparseInfer's prediction method into our DynamicInfer framework, replacing our MLP predictor while retaining our dynamic offloading strategy. The original SparseInfer paper compares against a version of PowerInfer where all neurons are placed on the GPU, which does not leverage PowerInfer's core CPU offloading feature. Our comparison is more direct.
>
> | Method | RTE | PIQA | COPA | Winogrande | Speed (token/s) |
> | :--- | :--- | :--- | :--- | :--- | :--- |
> | DynamicInfer+SparseInfer | 65.34 | 75.35 | 84.00 | 63.61 | 27.2 |
> | DynamicInfer (ours) | 68.47 | 77.47 | 87.00 | 65.96 | 24.4 |
>
> The results show that SparseInfer's predictor-less approach, by being lighter, allows more neurons to reside on the GPU, yielding a slight speed improvement. However, its lower prediction accuracy leads to a noticeable drop in task accuracy.
>
> This experiment demonstrates that our framework is adaptable to different predictors. The choice of predictor involves a trade-off between inference speed and model accuracy. We believe that developing more efficient and accurate predictors is a valuable research direction that is complementary to our core contribution.
>
> ---
>
> ### **Weakness 1.2: PowerInfer Implementation**
>
> > Section D.1 states that the open source implementation of PowerInfer differs from the main paper. The results are hence unfair to compare. Authors should ideally contact the authors to implement and/or discuss the implications to the results.
>
> We appreciate the reviewer's attention to this detail. There are two primary reasons for the performance discrepancy between our baseline results and those in the original PowerInfer paper:
> 1.  The open-source PowerInfer code we used lacks the fallback-to-dense mechanism for long input sequences, which the original paper mentions is used to optimize prefill.
> 2.  Our experimental hardware, particularly the CPU's single-core performance, is different from that used in the original work.
>
> Despite these differences, we believe our comparison is fair and meaningful. Our work, DynamicInfer, is built as an enhancement upon the same open-source PowerInfer implementation used as our baseline. Therefore, the performance gains we report are directly attributable to our proposed dynamic scheduling techniques, as all other system and hardware factors are held constant.

---

> ### Author Response · Authors · 2025-11-22
> **Author Response to Reviewer uAnA (2/4)**
>
> ### **Weakness 1.3: Latency Evaluation on Datasets**
>
> > Latency evaluation on at least one more dataset can help show robustness.
>
> We intentionally separated accuracy and speed evaluations because conducting latency tests on specific NLP datasets can produce misleading results. For prefill-heavy tasks, the test would primarily measure prefill speed. For decoding-heavy tasks, different methods generate a variable number of tokens, making a direct time comparison problematic. To ensure a fair and unambiguous evaluation of system performance, we measured raw generation speed (tokens/s) independently of specific dataset tasks.
>
> ---
>
> ###  **Weaknesses 2.1 & 2.3: Sparse Prediction Accuracy**
>
> > How accurate is the trained sparse predictor for new samples? across layers? across models?
> > How is k decided for the sparsity predictor beforehand? Is it data/hardware/model agnostic? what is the procedure?
>
> Sparsity predictors are model-specific and do not generalize across different model architectures, as activation patterns are unique to each model. Our predictor is trained on the C4 dataset and evaluated on its validation set.
>
> The table below shows the predictor's performance for ReluLlama-7B, varying the prediction distance `k` (i.e., predicting for layer `L+k` using activations from layer `L`).
>
> | Model | k | Predict Recall | RTE | PIQA | COPA | Winogrande | Speed (token/s) |
> | :--- | :-: | :---: | :--- | :--- | :--- | :--- | :--- |
> | ReluLlama 7B | 0 | 0.96 | 68.23 | 77.42 | 87.00 | 66.21 | 18.9 |
> | | 1 | 0.94 | 68.47 | 77.58 | 86.00 | 66.16 | 21.7 |
> | | **2** | **0.93** | **68.47** | **77.47** | **87.00** | **65.96** | **23.5** |
> | | 3 | 0.88 | 67.51 | 76.95 | 85.00 | 65.28 | 22.6 |
> | | 4 | 0.84 | 66.43 | 76.71 | 84.00 | 64.36 | 20.8 |
>
> Prediction recall decreases as `k` increases. We observe a significant drop for `k > 2`. Therefore, we selected `k=2` for our experiments, as it offers the best trade-off between prediction accuracy and the latency-hiding benefits of cross-layer prediction.
>
> ---
>
> ###  **Weakness 2.2: Constants Measurement**
>
> >  A number of constants are introduced w.r.t the optimization problem and constraints in E.3. How are they measured? How much do they depend on the hardware?
>
> These constants are determined through a brief, offline profiling process. They are highly hardware-dependent, reflecting the specific system's CPU-GPU I/O bandwidth and the computational speeds of the CPU and GPU. For a given hardware setup, these values can be treated as constants. We acknowledge this assumes a dedicated environment. In scenarios with resource contention from other applications, a more complex, online measurement approach would be required. Such a mechanism is not integrated into our current system but is a valuable direction for future work.
>
>
> ---
>
> ### **Weakness 2.3: Optimization Problem about Dynamic Sparsity Threshold**
>
> > E.4.1 is overly simplified. Lacks justification for Bernoulli distribution. “average error” can vary across layers and models.
>
> The Bernoulli distribution in our model pertains to the **activation status** of a neuron (i.e., whether it is active or not), rather than its specific floating-point value. The output of our MLP predictor for a given neuron is interpreted as the probability *P* of that neuron being activated. This probability is then used to model the neuron's behavior as a Bernoulli random variable. The primary motivation for this probabilistic modeling is to better utilize GPU resources by dynamically adjusting which neurons are offloaded, thereby minimizing GPU idle time while it waits for CPU computations.

---

> ### Author Response · Authors · 2025-11-22
> **Author Response to Reviewer uAnA (3/4)**
>
> ### **Weakness 2.4: Macro-level Explanation**
>
> > Can you explain the difference between your Macro-level residency constraint and PowerInfer? Although it makes sense to introduce, it does not seem very important from Fig10.
>
> The macro-level constraint captures sparsity at a semantic, sentence-level granularity, which lies between the token-level (micro) and dataset-level (static) views. This is based on the observation, also noted in prior work, that semantically similar inputs activate similar neural pathways.
>
> In our framework, the macro-level constraint is a crucial supplement to the micro-level (token-level) one. Relying solely on micro-level predictions would frequently suggest transferring more neurons than the CPU-GPU bandwidth allows. The macro-level constraint provides a stable, sentence-level filter, ensuring that a baseline of frequently used neurons for the current context resides on the GPU. This makes the dynamic transfer decisions more targeted and effective, preventing bandwidth saturation.
>
> ---
> ###  **Weakness 2.5: Greedy Algorithm Explanation**
> We regrettably acknowledge that finding the optimal solution in this context is extremely challenging. For each layer, every neuron has a binary choice of activation, resulting in $2^N$ possible combinations, where N represents the number of neurons in a single layer.. When considering the multiple layers of the model, it becomes computationally infeasible to directly determine the optimal solution.
>
>
>
> ---
>
> ###  **Weakness 2.6: Peak Memory Usage for GPU and CPU**
>
> > Can you provide a table showing peak memory usage for GPU and CPU for PowerInfer and DynamicInfer?
>
> The table below details the peak memory usage. DynamicInfer introduces only a minimal memory overhead compared to PowerInfer.
>
> | Model | Hardware | PowerInfer GPU (GB) | DynamicInfer GPU (GB) | PowerInfer CPU (GB) | DynamicInfer CPU (GB) |
> | :--- | :--- | :--- | :--- | :--- | :--- |
> | ReluLlama 7B | 2080ti | 11.3 | 11.2 | 7.4 | 11.8 |
> | ReluLlama 13B | 2080ti | 11.4 | 11.6 | 21.3 | 36.6 |
> | ReluLlama 13B | 4090 | 23.5 | 23.7 | 36.6 | 39.8 |
> | ReluLlama 70B int4 | 4090 | 23.6 | 23.5 | 24.2 | 28.4 |
>
> The additional CPU memory in DynamicInfer is primarily for the predictor and buffers for dynamic neuron transfer. The GPU memory footprint remains nearly identical.

---

> ### Author Response · Authors · 2025-11-22
> **Author Response to Reviewer uAnA (4/4)**
>
> ### **Question 1: Generality**
>
> > While the comparison with PowerInfer is reasonable, can you justify comparing your method against llama.cpp? It supports a much larger class of models (non-ReLU, quantization, etc) and was not tailored for efficient inference. It seems unfair.
>
> | Model | Method | RTE | PIQA | COPA | Winogrande | Speed (token/s) | Speedup |
> | :--- | :--- | :--- | :--- | :--- | :--- | :--- | :--- |
> | **Llama2-7B + 2080ti** | Llama.cpp | 63.90 | 78.07 | 89.00 | 69.38 | 7.9 | 1.00x |
> | | Powerinfer | 63.17 | 76.71 | 88.00 | 67.48 | 14.2 | 1.79x |
> | | DynamicInfer | 63.54 | 76.55 | 88.00 | 67.56 | 16.5 | 2.09x |
> | **Llama2-13B + 2080ti**| Llama.cpp | 67.15 | 79.43 | 91.00 | 72.45 | 2.5 | 1.00x |
> | | Powerinfer | 66.45 | 77.80 | 89.00 | 71.50 | 5.27 | 2.11x |
> | | DynamicInfer | 66.01 | 77.69 | 90.00 | 71.43 | 5.81 | 2.32x |
> | **Llama2-13B + 4090** | Llama.cpp | 67.15 | 79.43 | 91.00 | 72.45 | 9.8 | 1.00x |
> | | Powerinfer | 66.45 | 77.80 | 89.00 | 71.50 | 12.4 | 1.27x |
> | | DynamicInfer | 66.01 | 77.69 | 90.00 | 71.43 | 13.8 | 1.41x |
>
> To demonstrate the generality of our framework beyond ReLU models, we evaluated it on standard Llama-2 models. For non-ReLU models where true zeros are rare, we adapted the strategy from CATS [1], predicting and pruning a fraction of neurons with the lowest activation magnitudes. This experiment confirms that DynamicInfer's scheduling system is effective even with different definitions of "sparsity," highlighting its modularity.
>
> Furthermore, we view our method as **orthogonal** to other general optimizations like quantization and speculative decoding. DynamicInfer can be integrated with these techniques. For example, it can accelerate the draft model in speculative decoding or be applied to quantized models, as we demonstrated with the `ReluLlama-70B int4` model in our paper.
>
> > [1] Lee, Donghyun, et al. "CATS: Context-Aware Thresholding for Sparsity in Large Language Models." *First Conference on Language Modeling.*
>
> ---
>
> ### **Question 2: Sensitivity to λ**
>
> > How sensitive is the method w.r.t to λ, the neuron importance metric.
>
> The parameter λ balances the influence of micro-level (dynamic) and macro-level (semi-static) sparsity. We evaluated its sensitivity on ReluLlama-70B.
>
> | λ | Speed (token/s) |
> | :--- | :--- |
> | 0 (Micro-level only) | 6.41 |
> | **0.5 (Balanced)** | **6.71** |
> | 1 | 6.52 |
> | 2 | 6.27 |
> | ∞ (Macro-level only) | 5.36 |
> | PowerInfer (Baseline) | 5.14 |
>
> -   When `λ = 0`, the policy is purely dynamic but is limited by CPU-GPU bandwidth.
> -   When `λ = ∞`, the policy is purely semi-static (sentence-level), which is suboptimal as the set of "hot" neurons is not granular enough, leading to GPU underutilization.
>
> The results show that a balanced approach (`λ = 0.5`) yields the best performance by combining the strengths of both sparsity patterns.
>
> ---
>
> ###  **Question 3 & Weakness 3: Other Issues**
>
> > Feedback on writing and presentation.
>
> We thank the reviewer for the constructive feedback on the manuscript's presentation.
> -   For the second point, we opted to place the images at the top of each page, ensuring a balanced distribution across pages to enhance the visual aesthetics.
> -   Due to page limitations, we were compelled to relocate certain content to the appendix.
> -   The reference in Line 361 is a typo and should be "DynamicInfer." We will correct this.
> -   We will address all other noted writing issues during the revision process.

---

### Meta-Review · Area_Chair_6Kz6 · 2026-01-06

**Summary:**

The AC thinks that the initial version of the paper without rebuttal, is a reject because the reviewers pointed out many valid points.

After rebuttal, the AC thinks that the authors did a very good job addressing all the concerns.

A couple of things that the AC has in mind: (1) can we assume sparseness in all scenarios? (2) the writing needs to be improved.

Weighing the pros and cons, the AC decides to recommend an accept due to the rebuttal addressing almost all the concerns.

**Reviewer Concerns:**

All were addressed.

**Reviewer Scores:**

AC believes scores would have been raised after rebuttal.

---

### Decision · Program_Chairs · 2026-01-26

Accept (Poster)